# Inhaled Medicines for Targeting Non-Small Cell Lung Cancer

**DOI:** 10.3390/pharmaceutics15122777

**Published:** 2023-12-14

**Authors:** Arwa Omar Al Khatib, Mohamed El-Tanani, Hisham Al-Obaidi

**Affiliations:** 1School of Pharmacy, University of Reading, Reading RG6 6AD, UK; a.al-khatib@pgr.reading.ac.uk; 2Faculty of Pharmacy, Al Ahliyya Amman University, Amman 19111, Jordan; eltanani@rakmhsu.ac.ae; 3College of Pharmacy, RAK Medical and Health Sciences University, Ras Al Khaimah P.O. Box 11172, United Arab Emirates

**Keywords:** NSCLC, passive and active targeting, nanoparticles, immunotherapy, liposomes, inhalers

## Abstract

Throughout the years, considerable progress has been made in methods for delivering drugs directly to the lungs, which offers enhanced precision in targeting specific lung regions. Currently, for treatment of lung cancer, the prevalent routes for drug administration are oral and parenteral. These methods, while effective, often come with side effects including hair loss, nausea, vomiting, susceptibility to infections, and bleeding. Direct drug delivery to the lungs presents a range of advantages. Notably, it can significantly reduce or even eliminate these side effects and provide more accurate targeting of malignancies. This approach is especially beneficial for treating conditions like lung cancer and various respiratory diseases. However, the journey towards perfecting inhaled drug delivery systems has not been without its challenges, primarily due to the complex structure and functions of the respiratory tract. This comprehensive review will investigate delivery strategies that target lung cancer, specifically focusing on non-small-cell lung cancer (NSCLC)—a predominant variant of lung cancer. Within the scope of this review, active and passive targeting techniques are covered which highlight the roles of advanced tools like nanoparticles and lipid carriers. Furthermore, this review will shed light on the potential synergies of combining inhalation therapy with other treatment approaches, such as chemotherapy and immunotherapy. The goal is to determine how these combinations might amplify therapeutic results, optimizing patient outcomes and overall well-being.

## 1. Introduction

The respiratory tract has three major physiological functions: ventilation, diffusion, and perfusion. Ventilation is the process of breathing in which air enters the lungs due to the mechanical movement of the diaphragm. This process also involves air movement from the blood into the bronchial trees and alveolar space. Diffusion is the molecular mechanism that encompasses carbon dioxide transport, which ultimately affects the pH of the blood. Perfusion is critical for ventilation and diffusion as it maintains efficient blood circulation in order to match the high rate of gas exchange in the lungs [1].

The passage of air and the amount of energy required for ventilation is influenced by several physical factors [2]. These factors include alveolar surface tension, lung compliance, and airway resistance, which are vital for maintaining optimal respiratory function. One of the most important of these factors is alveolar surface tension, which refers to the tendency of the surface of the alveoli to resist expansion. This is reduced by lung surfactant, which coats the alveolar surface and decreases surface tension. By doing so, the surfactant helps keep the alveoli open during exhalation and promotes easier breathing during inhalation. Without surfactants, the airways would collapse after exhalation, making re-inflation during inhalation harder and less efficient.

Lung compliance is another crucial factor and refers to the ability of the lungs to expand when the air volume is increased. It is expressed as the volume change per unit of pressure change (mL/cm H_2_O or L/cm H_2_O) and depends on two essential factors: elasticity and surface tension. Elasticity refers to the tendency of the lung tissue to return to its original shape after being stretched, while surface tension refers to the tendency of the liquid lining the alveoli to resist expansion. Decreased surface tension and increased elasticity of the lung tissues enhance pulmonary compliance, allowing the lungs to rebound after being stretched during inhalation. This, in turn, increases the upstream pressure, which promotes better ventilation. Finally, airway resistance refers to the force that resists air flow through the airways. It is influenced by several factors, including the diameter and length of the airways, as well as the viscosity of the air. Higher airway resistance makes it harder for the air to move through the airways, leading to breathing difficulties. Understanding these physical factors is vital to maintaining healthy respiratory function and can help to identify and treat any underlying respiratory conditions.

## 2. Lung Cancer, Etiology, and Current Practice

Lung cancer remains a predominant cause of cancer-induced fatalities across the globe, representing nearly one-fourth of these tragic deaths [3]. Cancer is complex and therefore requires treatments that are dependent on several factors, such as the stage of the disease, its specific location within the body, the patient’s overall health and age, and other underlying medical conditions. Predominantly, localized cancers that have not spread beyond their point of origin are treated with surgical interventions, followed by chemotherapy sessions [4]. Non-small-cell lung cancer (NSCLC) encompasses a variety of lung cancer types, including adenocarcinoma, squamous cell, and large-cell undifferentiated carcinoma. Adenocarcinoma is the most prevalent, constituting about 40–50% of NSCLC cases [5], with bronchiolo-alveolar carcinoma accounting for 10–15% of these cases [6]. While squamous cell cancer typically originates at the central regions of the lung, large-cell undifferentiated carcinoma is characterized by its rapid proliferation and its ability to manifest anywhere within the lung [7]. Collectively, NSCLC represents roughly 85% of all diagnosed lung cancer cases, with a significant majority of patients being elderly. The average age of diagnosis is around 70 years [8].

Tobacco smoking remains the main cause of NSCLC, being responsible for nearly 90% of all cases, yet there are other contributors like exposure to second-hand smoke, the presence of radon, environmental pollution, genetic predispositions towards lung cancer, and certain genetic markers like the *CYP1A1* gene variant [9]. External factors, such as exposure to harmful substances like asbestos, tar, and specific metals, also play a pivotal role [10]. The probability of developing lung cancer is intrinsically tied to the frequency and longevity of smoking habits [11]. Interestingly, individuals with HIV present heightened risks of lung cancer as compared to the broader population [12], and a noticeable association has been established between pulmonary fibrosis and increased lung cancer susceptibility [13].

For NSCLC, the range of therapeutic alternatives includes surgery, radiation, chemotherapy, and specialized medical interventions. Chemotherapy typically involves a combination of a platinum-based compound with other therapeutic agents. First-line treatments usually encompass a platinum-based compound, synergized with a third-generation cytotoxic drug [14]. There are FDA-sanctioned treatments for NSCLCs exhibiting EGFR mutations, including compounds like gefitinib, erlotinib, and others [15]. Moreover, Bevacizumab, a compound targeting VEGF, has been approved for the first-line treatment of non-squamous NSCLC, especially when paired with chemotherapy [6]. Lung carcinoid tumors, although rare, are categorized as either typical or atypical carcinoids, both originating from neuroendocrine cells. Central carcinoids emerge within the central regions of the lungs, while peripheral carcinoids evolve toward the lungs’ extremities.

For patients with metastatic lung carcinoids, a variety of treatments is available, which range from surgical procedures to chemotherapy. Subcutaneous administrations of drugs like octreotide are commonly the go-to treatments, while targeted drugs like everolimus also show efficacy. To treat stage III atypical carcinoids, a mix of cisplatin and etoposide is typically used and is occasionally augmented with radiation sessions. The European Neuroendocrine Tumor Society has approved adjuvant therapy for atypical carcinoids with lymph node involvement, but such interventions are not recommended for typical carcinoids [16]. As medical research progresses, newer therapeutic strategies continue to emerge, offering hope and improved outcomes for patients globally. It is crucial to understand that each patient’s treatment plan is tailored based on their unique medical history and the specific nature of their cancer. Collaborations between oncologists, radiologists, and other medical professionals ensure the best care and outcome for each individual. In the area of cancer therapy, especially NSCLC, the development of drug delivery systems containing naturally occurring compounds has attracted attention. Recent research has described the anticancer effects of the natural cAMP-activating drug forskolin in NSCLC cells. It was shown that forskolin can inhibit both proliferation and migration in NSCLC cells. Interestingly, forskolin-mediated synergistic effects against Paclitaxel-induced cytotoxicity were also observed [17]. In a different study, the sustained release and intracellular accumulation of forskolin was found to prevent outer hair cells apoptosis [18].

## 3. Advantages of Direct Delivery to the Lungs to Target Lung Cancer

The administration of drugs to the lungs, or pulmonary delivery, has emerged as a highly promising option for drug delivery due to its numerous advantages. The lungs have a high surface area, which allows for the rapid and efficient absorption of drugs [19,20]. As a result, the respiratory tissues have the ability to absorb large quantities of drugs, making them an ideal route for drug delivery. Compared to the oral route, pulmonary drug delivery offers a faster onset of action, higher bioavailability, and better patient compliance, all while being non-invasive in nature [21,22,23,24]. Moreover, inhaled medication does not undergo first-pass metabolism, leading to lower drug degradation. This method also promotes the direct effect of the drug on the lungs, resulting in better selectivity and reduced circulation into the body’s system, which in turn leads to fewer side effects and better therapeutic outcomes.

Studies have demonstrated that inhaling anticancer agents can effectively combat micro metastasis and distant metastasis by targeting the lymphatic drainage of the alveoli [25]. The interstitial space is crucial in delivering drugs directly to the lungs, which is vital for suppressing lung cancer metastasis [19]. Upon being excreted from the lungs, these drugs can accumulate in the interstitium, the peripheral airways, and the pulmonary lymphatic system, enabling them to interact with cancer cells beyond the alveolar space and to restrict the invasiveness and metastasis of cancer [19]. Pulmonary drug delivery also limits systemic exposure to drugs [26]. Intravenous treatments are impeded by poor selectivity, systemic toxicity (including nausea, vomiting, nephrotoxicity, neurotoxicity, and anaemia), and the development of multidrug-resistant pumps [26,27,28], as well as the necessity for optimising drug dosing [29].

## 4. Lung Surfactants and Impact on Drug Deposition 

Lung surfactants, also known as endogenous surfactants, are a combination of lipids and proteins found in the lung’s alveolar lining. Their main role is to decrease surface tension by forming a single layer at the alveolar air-liquid boundary [30]. Lung surfactants facilitate the efficient spreading of drugs delivered to the lungs across mucus surfaces and improve the aerosol transport between lung areas, resulting in more consistent drug dosing within the deeper lung regions.

Exogenous surfactants are employed when there is a surfactant deficiency, serving as a substitute treatment for respiratory distress syndromes [31]. These can be either natural or artificially produced. Natural surfactants primarily contain phospholipids, mainly dipalmitoylphosphatidylcholine (DPPC), neutral fats (chiefly cholesterol), and proteins. Frequently used natural surfactants come from sources like humans, animals, cows, and pigs. On the other hand, synthetic exogenous lung surfactants like colfosceril and lucinactant are entirely synthesized, often based on DPPC.

The surfactant layer plays a crucial role in drug deposition within the lungs [32]. Inhaled drugs must first navigate this layer to access the underlying tissue. The presence of lung surfactants influences drug deposition in multiple ways, such as:Diffusion limitation: The surfactant layer might act as a barrier, hindering drugs from reaching deeper tissues. This can lead to reduced drug concentrations in the lungs and diminished treatment effectiveness.Extended residence time: The surfactant layer can prolong the duration of time that the drugs linger in the lungs. This is beneficial for drugs that are rapidly expelled, as it allows for extended exposure to lung tissue, which enhances their potency.Drug particle dimensions: The size of the drug particles affects their deposition depth in the lungs. Larger particles might be trapped atop the surfactant layer, which limits their penetration, while smaller particles might navigate past the layer to access the deeper lung regions.

In essence, lung surfactants can enhance deposition by ensuring even alveolar expansion during inhalation and enhancing alveolar resistance to collapse upon exhalation [33]. Additionally, lung surfactants quickly adhere and redistribute effectively at the air–water boundary, facilitating the even spread of inhaled drugs across the mucus-covered lung surfaces, optimizing drug deposition [30]. Thus, the surfactant quantity present can considerably influence the drug amounts deposited in the lungs.

## 5. Mucociliary Clearance and Mucoadhesion, Getting the Balance Right

Mucociliary clearance (MCC) represents the mechanism by which the respiratory system expels foreign entities, such as medications, via the synchronized efforts of cilia and mucus. It is an essential protective function that safeguards the lungs from inhaled invaders like allergens, harmful substances, and microbes [34]. This protective action propels the mucus, along with any dissolved or adhered substances, from the lung’s tracheobronchial region to the GIT. Conversely, the alveolo-bronchiolar clearance process shifts solutes, lining fluid, and/or macrophages from the alveoli towards the MCC area [35]. An extended clearance phase, lasting several days to several weeks, is characterized as the peripheral and alveolar removal of particles via the macrophages. There is a notable correlation between how long particles stay in the airways and their physical dimension. MCC quickly expels larger particles from the airways, while tinier particles remain longer [36]. It has been suggested, however, that the MCC’s efficiency diminishes in the deeper sections of the respiratory system [35].

Multiple factors can modulate the MCC, including those influencing the mucus or cilia. Factors like augmented mucus secretion, reduced mucus thickness, and elevated ciliary motion can enhance the MCC’s speed. In contrast, altering the unique properties of mucus or disrupting its structure can decelerate the MCC. Ambient conditions, such as temperatures around 23 °C, can induce a moderate decline in MCC rates [37]. Additionally, smoking can hinder the MCC by reducing ciliary abundance, thereby altering mucus dynamics. Moreover, various upper respiratory ailments can affect the MCC due to their effects on ciliary motion and/or mucus behavior; these include conditions like asthma, rhinitis, and chronic sinusitis [37]. Accelerated MCC is seen as a limitation for inhaled treatments as it limits the time that drugs remain at the deposition locale, thereby lowering the drug bioavailability [36,38]. To counter this challenge, there is a need to enhance mucoadhesion, which signifies a drug’s capability to bind to the mucus layer and ensure prolonged residence in the lungs [37]. Utilizing innovative mucoadhesive drug delivery systems capable of adhering to the pulmonary mucosa [39] will extend the retention period at the action site [40,41] and enhance the interaction between the pulmonary mucosa and the medication, ensuring a broader drug spread. This mucoadhesive approach elevates drug concentrations at the deposition site and increases drug uptake, thereby elevating drug bioavailability and enhancing its absorption [42,43]. In essence, both mucociliary clearance and mucoadhesion are essential determinants that influence drug transport within the respiratory system. Achieving the right equilibrium between these elements is paramount for ideal respiratory drug delivery. While it is crucial to reduce the MCC to boost the performance of drugs delivered to the lungs via mucoadhesive techniques, this adjustment should be performed carefully, as the MCC acts as a native defense strategy of the lower respiratory tract against potential infections, and overly aggressive mucoadhesive actions could jeopardize this defense [44]. Overall, the MCC can positively affect the efficacy of inhaled anticancer therapies because the MCC is crucial for removing inhaled particles including anticancer therapeutic agents, which prevents drug accumulation and reduces toxicity. The MCC can also affect the distribution of drugs across the respiratory tract. Thus, understanding the MCC is essential for designing inhalable drug formulations that can target lung cancer effectively. On the other hand, The MCC poses additional challenges because the rapid clearance of inhaled anticancer therapies can limit the time that a therapeutic agent spends in target tissue, especially within the lungs. Therefore, optimizing drug formulations for effective penetration is crucial for balancing the contact time with the mucosal surface absorption risk, as diseases affecting mucociliary clearance vary.

## 6. Methods and Devices for Inhalation

Inhalation devices are commonly utilized for administering medications in the form of finely-ground particles, suspensions, or solutions. The efficacy of drug delivery to the lower respiratory tract is influenced by a multitude of factors, including the type of inhaler used, the internal resistance of dry powder inhalers, the drug formulation, the size of the particle, the velocity of the produced aerosol of pressurized metered-dose inhalers, and the ease of use [45]. It should be noted that each inhalation device has its own unique set of advantages and disadvantages, which can affect the way that medication is delivered to the lungs. Consequently, it is essential to carefully consider the characteristics of each inhaler before selecting an appropriate device for drug administration [46].

The most commonly used devices for inhalation include pressurized metered-dose inhalers (PMDIs), which are inhalers that use a metering system to correctly dispense the desired propellant volume; PMDIs with spacers or valves holding chambers that serve as aerosol reservoirs and are linked to a pressurized metered-dose inhaler. These devices are especially beneficial for small children because they require less coordination between inhaling and dispensing medication. Additionally, there are the breath-actuated pressurized metered-dose inhalers that are activated by the patient’s inhalation, removing the requirement for hand-to-breath coordination [47]. One of the most essential advantages of these inhalers is their ability to administer several doses of medication. The dry powder inhalers (DPIs) also spread medication as a powdered substance of respirable-sized micronised particles. Moreover, there are nebulisers which generate aerosol particles by either using a high-velocity gas jet of oxygen or air (jet nebuliser) or a rapidly vibrating piezoelectric crystal (ultrasonic nebuliser) [47]. Soft mist inhalers are types of nebulizers that are designed to convert liquid medicine into inhalable droplets, similar to handheld nebulizers and multi-dose devices like metered-dose inhalers and dry powder inhalers [47].

## 7. Assessment of Drug Deposition, Current Methods, and Accuracy for Assessment of Deposition of Anticancer Agents

Accurately assessing the distribution of drugs throughout the lungs is crucial when evaluating their efficacy in treating lung cancer via inhaled chemotherapy [48]. Administering the appropriate chemotherapeutic agents at the correct doses, intervals, and disease locations can lead to better outcomes for patients [49]. Factors such as particle size, density, shape, velocity, charge, hygroscopicity, and surface properties [50], as well as the patient’s breathing pattern and respiratory system condition, are all vital factors for efficiently depositing the particles in the lungs [51]. Various methods, including lung imaging techniques, pharmacokinetic analysis, and direct drug concentration assay in lung tissue, can be used to assess the deposition of anticancer agents [52]. However, the accuracy of these methods depends on the types of anticancer agent, delivery system, and target tissue. The process of determining the rate of the absorption of inhaled drugs is crucial for determining their effectiveness [53].

When it comes to assessing the absorption and disposition of inhaled medications meant for systemic administration, there are different methods that can be used. These methods include in vivo, in vitro, and ex vivo models. The Andersen cascade impactor is a commonly used method for determining lung deposition. The next generation impactor (NGI) is another method that calculates various aerosolization performance parameters such as mass median aerodynamic diameter (MMAD), geometric standard deviation (GSD), fine particle fraction (FPF), emitted fraction percent (EF%), and residual fraction percent (RF%) based on the drug deposition percentage at each step [54]. On the other hand, the isolated perfused lung (IPL) is an ex vivo technique that involves removing the lungs from an animal, such as a rodent or guinea pig, and placing them in a synthetic environment.

The advancement of inhaled therapeutics and the exploration of aerosol dynamics have significantly benefited from in silico techniques, which robust mechanisms for anticipating deposition within respiratory tracts [55]. Numerous computational strategies, intricate airway designs, and diverse suppositions related to flow and aerosol mechanics have been employed in past numerical examinations. An array of in silico techniques exists for forecasting the aerosol accumulation in the upper respiratory tracts, encompassing medical imaging-derived airway designs from techniques like computed tomography (CT) and magnetic resonance imaging (MRI), airflow designs such as Reynolds-averaged Navier-Stokes (RANS), large eddy simulation (LES), and direct numerical simulation (DNS), alongside aerosol movement and accumulation models like Lagrangian or Eulerian methodologies [56].

The in vivo-in vitro correlation (IVIVC) procedure plays a crucial role in pharmaceutical advancement, linking the deposition of medicines in human lungs to estimates derived from in vitro physical respiratory designs and in silico computational models. This correlation offers insightful projections concerning a drug’s final destination post-administration through diverse inhalation aerosol mechanisms. The process of labelling is integral in this context, necessitating validation to ascertain the radiotracer’s accuracy as a drug marker prior to the radioactive aerosol’s inhalation by the patient or participant. This entails quantifying the dosage and particle size distribution of the aerosolized medication and radiotracer and comparing these findings with the unlabelled drug [57].

Gamma scintigraphy is a widely used method for evaluating lung deposition accuracy and is often used to quantify whole-lung deposition and oropharyngeal deposition. However, despite its effectiveness, single-photon emission computed tomography (SPECT) and positron emission tomography (PET) have a stronger theoretical basis for quantification [58]. The multi-planar information provided by SPECT and PET enables a precise and in-depth evaluation of the deposited dosage in various lung regions [59]. Overall, the accuracy of these methods can vary depending on various factors, including the sensitivity and specificity of the imaging technique, the timing and duration of drug administration, and the variability in patient anatomy and breathing patterns.

## 8. Effect of Particle Size and Shape on Their Lung Deposition

As the inhaled particles travel through the respiratory tract, their physical characteristics including size, shape, and density influence their inertia and, consequently, their deposition [50,60,61]. This behaviour is largely defined by the particles’ aerodynamic diameters, which correspond to the diameter of a unit-density sphere that achieves the same velocity in the airstream as the particle in question. Techniques such as light scattering, laser diffraction, or image analysis can be employed to measure the geometric diameter of these particles, which can then be translated into their aerodynamic diameters [62].

The deposition of these inhaled particles in the airways occurs primarily through three mechanisms: inertial impaction, gravitational sedimentation, and diffusion [63,64]. Particles larger than 5 µm generally cannot alter their path within the airflow and tend to impact and deposit in the upper airways through inertial impaction [65,66,67]. Particles between 1 and 5 µm usually settle in the lower airways (bronchioles and alveoli) through gravitational sedimentation [68]. Meanwhile, particles smaller than 1 µm often stay suspended in the airstream and are likely to be exhaled without depositing, with diffusion being their main deposition mechanism [68,69]. Notably, particles smaller than about 500 nm may show increased lung deposition [70,71,72].

For inhaled chemotherapeutic agents, particles within the aerodynamic diameter range of 1–5 µm are preferred [73]. The effectiveness of these inhaled formulations is frequently assessed based on the fine particle fraction (FPF) or fine particle dose (FPD), which refers to the portion or dose of particles within this size range. Another important metric is the mass median aerodynamic diameter (MMAD), the diameter below which 50% of the particles fall, which serves as an indicator of the aerosol characteristics of these formulations [60]. Also, the clearance of the inhaled particles is highly dependent on their size and occurs through three primary mechanisms: mucociliary clearance, phagocytosis, and systemic uptake [74].

Mucociliary clearance is the most prominent mechanism in the upper airways [75]. Here, mucus secreted by the ciliated columnar epithelium traps particles, which are then moved upward by cilia and eventually coughed out or swallowed This method is particularly effective for particles larger than 5 μm [76]. In contrast, smaller particles, which reach the deeper parts of the lungs, remain there longer as mucociliary clearance is less dominant in these regions [77,78]. While macrophages are present in the upper airways, their role in phagocytosis is less pronounced [79].

In the deeper lungs, the clearance mechanisms are more complex and largely depend on the particles’ dissolution characteristics. Slowly dissolving or insoluble particles can be cleared through a combination of mucociliary clearance, phagocytosis by alveolar macrophages, and endocytosis [80,81,82]. Alveolar macrophages predominantly handle the clearance in the deep lungs, dealing mainly with particles sized between 1 and 5 μm [83]. These macrophages engulf the particles, which are then either digested lysosomally or expelled through the lymphatic system or via mucociliary clearance [83]. Particles smaller than 200 nm often evade detection by macrophages due to their size. They can be quickly taken up by the epithelial cells or translocate to the systemic circulation through protein/receptor-mediated mechanisms or by endocytosis via alveolar caveolae [84].

Another important aspect affecting the lung drug delivery of inhaled chemotherapy is the shape of the inhaled agents. The larger the contact areas, such as elongated shapes, the less effective inhalation is for lung targeting. This is because their shape leads to stronger van der Waals forces, resulting in a higher tendency for the particles to clump together or aggregate [85]. This was investigated by a study that compared particles that had similar sizes but different shapes, including spheres, cubes, plates, needles, and pollen-like forms. This study aimed to explore how particle shape affects the particles’ movements, aerosolization, and ability to deposit in the lungs. It was found that particles resembling pollen were most advantageous for inhalation drug delivery. Compared to the other shapes, these pollen-like particles demonstrated superior aerosolization, flowability, and deposition qualities, making them more suitable for this application [86].

The efficacy and safety of inhaled anti-lung cancer treatments depend on the drug deposition in the lungs. Several factors determine the success of these therapies. The site of action is critical because the medication targets tumor cells inside the lungs. Therefore, it is crucial for drug particles to deposit at the target site for therapeutic efficacy. The size and dispersion of the medication particles determine their deposition in the respiratory tract. Ideally, the particles should be small enough to reach the deeper areas of the lungs where cancer cells reside. The aerodynamic properties of the particles significantly impact their penetration and deposition. Additionally, individual differences in lung function and physiology can influence inhaled drug deposition patterns. Factors such as respiratory rate, tidal volume, and airway geometry can alter drug deposition. The appropriate device selection should be based on the drug and patient’s specific properties and the formulation of the anti-cancer agent, which influences its deposition in the lungs. Finally, the drug’s distribution in the tumor microenvironment is critical and can be affected by factors such as blood flow, tumor vascularization, and interstitial pressure.

## 9. Challenges for the Delivery of Inhaled Chemotherapy

Inhaled therapy is a popular method for treating respiratory disorders like asthma, chronic obstructive pulmonary disease (COPD), and cystic fibrosis. This method delivers medication straight to the lungs through inhalation. Nevertheless, various challenges come with inhaled chemotherapy that can affect its safety and effectiveness.

### 9.1. Uniform Drug Deposition

It is difficult to achieve the homogenous deposition of the chemotherapeutic medication in the lungs. Inhaled medications may not be distributed equally throughout the respiratory system, resulting in drug concentration differences at distinct lung areas.

### 9.2. Patient Variability

Variability in patient anatomy, respiratory rates, and inhalation patterns can all have an impact on inhaled chemotherapeutic deposition. It is difficult to provide consistent and effective drug distribution across a heterogeneous patient group.

### 9.3. Device Design and Performance

Inhalation device design and function are critical for medicine delivery. Issues such as device blockage, poor aerosolization, or insufficient patient breathing techniques can all have an impact on inhaled anti-lung-cancer agents’ delivery efficiency.

### 9.4. Disease-Specific Challenges

Inhaled chemotherapy may bring distinct complications for certain lung conditions. Conditions such as chronic obstructive pulmonary disease (COPD) or cystic fibrosis, for example, may change lung physiology, which in turn impacts drug deposition and effectiveness.

### 9.5. Toxicity and Side Effects

Chemotherapy inhalation may induce local irritation or systemic side effects. The continuous difficulty of designing inhaled chemotherapy formulations involves balancing therapeutic efficacy with minimizing side effects.

### 9.6. Drug Stability

Some chemotherapy medications could be affected by environmental conditions such as temperature and humidity. It is critical to ensure a drug’s stability during storage and inhalation in order to preserve its effectiveness.

## 10. Formulation of Anticancer Agents Using Carrier Free Technology

Carrier-free technology is an innovative approach to drug delivery and involves the direct transportation of the drug to the desired target without the need for any auxiliary carrier particles. This method has gathered significant attention due to its inherent ability to enhance therapeutic results while simultaneously reducing potential side effects [87]. In the area of nanotechnology, there has been a surge of comprehensive research that focuses on the delivery of anti-cancer drugs. The primary objective of these studies is to strengthen therapeutic effectiveness while limiting associated toxicities [88,89,90]. As the medical field continues to advance, there has been a notable evolution in the formulation of carrier-free nanodrugs. These nano-drugs are self-assembled using prodrugs, unmodified drugs, or amphiphilic drug-drug conjugates [91]. Their unique attributes, such as enhanced pharmacodynamics/pharmacokinetics, diminished toxicity, and superior drug-loading capabilities, have made carrier-free nanodrugs a focal point for oncological treatments. An essential area of exploration is the design and application of multi-functional carrier-free nanodrugs, especially those with potent anti-cancer properties suitable for clinical administration. In comparison to nanodrugs that utilize inorganic or organic carriers, carrier-free nanodrugs offer a plethora of advantages. Among those advantages is their impressive drug-carrying capacities combined with their reduced side effects [92,93,94]. Their engineered structures enhance accumulation within tumors [95,96]. Furthermore, the synthesis process of these carrier-free nanodrugs is both simple and environmentally friendly [95], often avoiding the use of harmful reagents like organic solvents. These nanodrugs also demonstrate prolonged retention in the bloodstream as compared to standard anti-cancer drugs [93,94]. This extended retention facilitates the build-up of anti-cancer agents within tumor tissues, leading to increased absorption by tumor cells [93]. Carrier-free technology stands out as a groundbreaking advancement in drug delivery, offering a multitude of benefits that are not present in the traditional carrier-based systems [97].

A variety of techniques have been innovatively developed for the synthesis of carrier-free nanodrugs. These include methods such as nanoprecipitation, template-assisted nanoprecipitation, thin-film hydration, the spray-drying approach, the supercritical fluid (SCF) method, and wet media grinding [98]. These techniques have catalyzed the latest breakthroughs in the domain of carrier-free nanodrugs tailored for oncological interventions. Broadly, these advancements can be segmented into three distinct categories:Self-assembly of a singular anti-cancer drug.Self-assembly of multiple anti-cancer drugs.Self-assembly of anti-cancer drugs in conjunction with other therapeutic agents, including photosensitizers, photothermal agents, immune reagents, and genetic materials [99,100].

The formulation of carrier-free nanodrugs often involves the strategic coupling of two drug entities or the conjugation of functional organic compounds—such as fatty acids, vitamins, proteins, and photosensitizers—to drug molecules. This is achieved through the establishment of covalent bonds and/or physical interactions. Such an intricate design enhances the stability, targeting precision, and therapeutic potency of anti-cancer medications [101]. In a research study, a unique carrier-free multidrug nanocrystal was employed for combination chemotherapy. This involved the assembly of three primary water-insoluble drugs into nanorods, which were subsequently conjugated with PEG to bolster their environmental stability [96]. Both the in vivo and in vitro assessments revealed that these multidrug nanocrystals exhibited a therapeutic efficacy that was up to three times superior to that of the unbound drugs, even when administered in equivalent dosages. Furthermore, they effectively overcame the phenomenon of multidrug resistance. In another study, distinct formulations of pure nanodrugs were prepared with curcumin and 10-hydroxycamptothecin, using a variety of solvents and anti-solvents [102]. As per the documented findings, the synthesis of the nanoparticles from agents like doxorubicin, ionidamine, and triphenylphosphine had the potential to diminish metabolic energy by specifically targeting the mitochondria [103]. Notably, these prepared nanoparticles succeeded in extending the half-life of doxorubicin in the circulatory system to a commendable 3 h [94].

A recent research study has made a significant breakthrough in creating lactose-free dry powder formulations of two important medicines, fluticasone propionate and salmeterol xinafoate FP/SX, by utilizing advanced organic solutions and a spray-drying particle engineering design. These formulations also feature mannitol as an excipient, which helps in molecular combination. The resultant spray-dried powders exhibit crucial properties that are essential for the effective administration of the inhaled medication [104].

Two other studies have reported on the development of carrier-free chemo-photodynamic nanodrugs. In one study, chlorine e6 and doxorubicin molecules self-assembled into nanoparticles through electrostatic π-π stacking and hydrophobic interactions [95]. The other study created a carrier-free nanodrug by using curcumin with a donor-acceptor pair composed of perylene and 5,10,15,20-tetro(4-pyridyl)porphyrin (H2TPyP) as photosensitizers [105]. This allowed for photodynamic therapy and curcumin inhibited cancer cell growth. The fluorescence state of the curcumin was only activated when it was released from the nano-platform, allowing for the self-monitoring of the drug release [94]. Combining more than two different drug types is often necessary to achieve a more effective therapeutic impact. Optimising the ratios of various chemotherapy drugs in nanodrugs will increase their efficiency while reducing the likelihood of additional side effects. Recently, different targeting ligands, such as antibodies, peptides, and specific cell membranes with a high affinity for cancer cells, have been developed to improve the targeting capability of carrier-free nanodrugs. To increase the stability and therapeutic effectiveness of carrier-free nanodrugs, which are somewhat unstable and prone to precipitation and aggregation, it is essential to incorporate amphiphilic protein, peptide, and nucleic acid preparations with therapeutic functions as stabilisers. Coating the carrier-free nanodrugs with different cell membranes is another promising approach for endowing the unprotected nanodrugs with excellent biocompatibility and biological stabilisation [106].

Two research studies have investigated the formulation of carrier-free chemo-photodynamic nanodrugs. The first study explored the self-assembly of chlorine e6 and doxorubicin molecules into nanoparticles. This assembly was facilitated by the intricate interplay of electrostatic π-π stacking and hydrophobic forces [95]. In a parallel study, a distinctive carrier-free nanodrug was prepared by combining curcumin with a donor-acceptor, specifically perylene and 5,10,15,20-tetro(4-pyridyl)porphyrin (H2TPyP), which acted as photosensitizers [105]. This innovative combination not only enabled photodynamic therapy but also effectively reduced the proliferation of the cancer cells. Interestingly, the fluorescence state of the curcumin was exclusively activated upon its release from the nano-platform, offering a unique mechanism for the real-time monitoring of the drug dispersion [94]. In cancer therapy, the combination of multiple drug variants is often imperative for achieving an enhanced therapeutic outcome. By optimising the proportions of diverse chemotherapy agents within nanodrugs, one can potentially increase their efficacy while simultaneously diminishing the risk of supplementary adverse reactions. In recent times, a plethora of targeting ligands, encompassing antibodies, peptides, and specific cell membranes with a pronounced affinity for malignant cells, have been innovatively designed. These advancements aim to augment the targeting precision of carrier-free nanodrugs. However, it is crucial to acknowledge that carrier-free nanodrugs, in their native states, exhibit certain limitations such as instability and tendencies for precipitation and aggregation. To counteract these challenges and improve the stability and therapeutic efficacy of these nanodrugs, the integration of amphiphilic proteins, peptides, and nucleic acid formulations with therapeutic attributes has been deemed essential. These components act as stabilizers, ensuring the robustness of the nanodrugs. Furthermore, enveloping the carrier-free nanodrugs with diverse cell membranes has emerged as a promising strategy. This encapsulation not only confers the nanodrugs with superior biocompatibility but also ensures their biological stabilisation, leading to enhanced therapeutic outcomes [106].

## 11. Strategies for Drug Targeting of NSCLC

Although various treatment options are available for NSCLC, the survival rate remains relatively low; therefore, there is a need for new treatment strategies. Figure 1 shows some examples of the current strategies for targeting NSCLC while Table 1 shows examples of the current pulmonary-delivered medicines that have been investigated for lung cancer.

### 11.1. Active Targeting for NSCLC in Inhaled Therapies

Active targeting is a therapeutic strategy that uses monoclonal antibodies or small-molecule inhibitors to specifically target the proteins or signaling pathways that are dysregulated in cancer cells. This technique involves the use of ligands or antibodies to selectively deliver therapeutic agents to the tumor site. One of the most commonly used approaches of active targeting is the targeting of overexpressed receptors on cancer cells, such as the epidermal growth factor receptor (EGFR) or the human epidermal growth factor receptor 2 (HER2). This method has been found to be effective in several cancer types, including non-small-cell lung cancer (NSCLC).

In NSCLC, targeted therapies have been developed that specifically target these receptors, including tyrosine kinase inhibitors (TKIs) and monoclonal antibodies (mAbs). For instance, osimertinib, a third-generation EGFR-TKI, has shown efficacy for treating NSCLC patients with EGFR mutations [107]. This drug has been found to be particularly effective in patients who have developed resistance to other EGFR-TKIs. Similarly, Trastuzumab, a mAb that targets HER2, has shown efficacy in NSCLC patients with HER2 overexpression [108]. Other examples of active targeting strategies for NSCLC include the use of ligands that target tumor-specific antigens, such as folate or transferrin receptors, or the use of nanoparticles that can selectively deliver therapeutic agents to the tumor site. These strategies are still in experimental phases and require further research and development. Active targeting is a promising therapeutic strategy for the treatment of NSCLC and other cancers. By selectively delivering therapeutic agents to the tumor site, this technique can minimize the side effects associated with traditional chemotherapy and improve patient outcomes.

#### 11.1.1. Immune Checkpoint Inhibitors (ICIs)

Immunotherapy has emerged as an effective treatment option for cancer patients. Immune checkpoint inhibitors (ICIs) are a type of immunotherapy that work by blocking specific pathways in the immune system. These pathways, known as immune checkpoints, are responsible for regulating the immune system’s response against cancer cells. One of the most important immune checkpoint pathways is the programmed death 1 (PD-1)/programmed death ligand 1 (PD-L1) pathway. Cancer cells can activate this pathway to evade detection by the immune system. ICIs such as pembrolizumab, nivolumab, and atezolizumab target the PD-1/PD-L1 pathway, preventing cancer cells from evading the immune system [109]. Another important immune checkpoint pathway is the cytotoxic T-lymphocyte-associated protein 4 (CTLA-4) pathway. This pathway regulates the activity of T cells, which play a crucial role in the immune response against cancer cells. ICIs such as ipilimumab target the CTLA-4 pathway, enhancing the activity of T cells against cancer cells. Non-small-cell lung cancer (NSCLC) is one type of cancer that can be treated with ICIs. These treatments have been shown to be effective in improving the survival rates and the quality of life for patients with NSCLC. Ongoing research aims to identify new ICIs and optimize their use in order to further improve outcomes for cancer patients.

#### 11.1.2. Tyrosine Kinase Inhibitors (TKIs)

Non-small-cell lung cancer (NSCLC) is often associated with mutations in receptor tyrosine kinases, such as EGFR and ALK. Tyrosine kinase inhibitors (TKIs) are a class of drug that have been developed to specifically target these mutations in the tyrosine kinase domain of these receptors. TKIs work by obstructing the downstream signaling pathways that are triggered by these mutations. By doing so, they prevent the cancer cells from proliferating and cause them to undergo apoptosis, or programmed cell death. This mechanism makes TKIs a promising treatment option for NSCLC. Several TKIs have been approved for the treatment of NSCLC, including erlotinib, afatinib, and crizotinib [110]. These drugs have shown significant efficacy in improving the overall survival and the quality of life of NSCLC patients. However, the choice of TKI may depend on the specific genetic profile of the tumor and the patient’s overall health condition. Overall, TKIs are a targeted therapy that offers a promising treatment option for NSCLC patients with mutations in their receptor tyrosine kinases. With the availability of several approved TKIs and ongoing research in this field, the future of NSCLC treatment looks more hopeful than ever before.

#### 11.1.3. Antibody-Drug Conjugates (ADCs)

Antibody-drug conjugates (ADCs) are a type of targeted cancer therapy that combines an antibody with a cytotoxic medication. The antibody is designed to specifically recognize and bind to a particular antigen on the surface of the cancer cells. Once the antibody attaches to the cancer cells, the cytotoxic medicine is released and delivered directly to the cancer cells, sparing the healthy cells from damage. This targeted approach can potentially enhance the efficacy of the chemotherapy while reducing its side effects. Rovalpituzumab tesirine and sacituzumab govitecan are some of the ADCs that are currently being studied for non-small-cell lung cancer (NSCLC) and have shown promising results in clinical trials [111].

#### 11.1.4. Personalized Inhaled Medicine Approaches

Individualizing inhaled medicines based on the genetic and molecular profiles of NSCLC tumors can improve treatment outcomes. Identifying specific mutations or indicators that can be targeted with personalized medicine formulations could be part of this approach.

### 11.2. Passive Targeting for NSCLC in Inhaled Therapies

Passive targeting is a method of delivering medicines to a tumor site by taking advantage of the increased permeability and retention (EPR) effect, which occurs due to the abnormal blood vessels in the tumor. This approach allows for a higher concentration of medicine to accumulate in the tumor tissues than in healthy tissues. Some examples of passive targeting strategies for NSCLC (Non-small-cell lung cancer) are listed below.

#### 11.2.1. Use of Amorphous Solid Dispersions

Solid dispersions refer to a formulation technique that involves dispersing a drug in a solid matrix. In this technique, the drug is mixed with an inert carrier molecule, which helps to reduce the loss of the drug within the inhalation device and minimize the deposition of the drug in the oropharyngeal area [112].

#### 11.2.2. Use of Nanoparticle Delivery Systems

Non-small-cell lung cancer (NSCLC) is a challenging disease to treat due to its complex molecular heterogeneity and resistance to conventional therapies. However, nanoparticle-based drug delivery systems have emerged as promising approaches for combatting NSCLC. These tiny particles offer numerous advantages, including the ability to selectively target cancer cells, enhance drug efficacy, and reduce toxicity. Inhalable self-assembled albumin nanoparticles conjugated with doxorubicin and octyl aldehyde and adsorbed with the apoptotic TRAIL protein represent a novel inhalation-based combination therapy method for the treatment of resistant lung cancer [113]. A study reported that inhaling mesoporous silica nanoparticles (MSNs) resulted in the preferential concentration of nanoparticles in mouse lungs, preventing the MSNs’ escape into the systemic circulation and limiting their accumulation in other organs. The experimental data showed that the proposed DDS meets the key requirements for effective non-small-cell lung cancer treatment [114].

#### 11.2.3. Lipid-Based Nanoparticle Delivery Systems (LNPs)

Lipid-drug conjugate nanoparticles are a type of drug delivery system that shows great potential for the targeted delivery of drugs to cancer cells, including NSCLC. These nanoparticles are composed of a core of drug molecules that are covalently attached to lipid molecules, forming a protective shell around the drug core. The lipid shells of these nanoparticles can be designed to be either hydrophilic or hydrophobic, enabling them to interact with different types of cells and tissues in the body and providing a platform for efficient drug delivery. The covalent linkage of drugs to lipid molecules allows them to self-assemble into nanoparticles, which can provideenhanced stability and circulation times as compared to other lipid nanoparticle formulations. The conjugation of the drugs to lipids also offers improved biocompatibility, making these nanoparticles safe and effective drug carriers.

A study found that the NLC was highly effective for the tumor-targeted inhaled delivery of anticancer drugs (doxorubicin or paclitaxel) and the delivery of siRNA mixtures specifically to lung cancer cells, resulting in efficient tumor growth suppression and the prevention of adverse side effects on healthy organs [115]. This technology has the potential to revolutionize cancer treatment by enabling targeted drug delivery and improving treatment outcomes.

#### 11.2.4. Liposomes

Liposomes are small, spherical structures made up of a double layer of phospholipids that can hold hydrophobic drugs within their core. They consist of one or more layers of lipid bilayer that encircle an aqueous pocket. Liposomes have been found to accumulate in non-small-cell lung cancer (NSCLC) tumors and can enhance the effectiveness of chemotherapy drugs in this type of cancer [116]. Several studies have demonstrated that liposomal formulations of chemotherapy drugs, such as the inhalable liposomal curcumin formulation, represent a promising pulmonary medicine for the treatment of lung cancer [117]. Another study showed that liposomal nanoparticles loaded with paclitaxel showed superior antitumor activity against NSCLC in vitro and in vivo as compared to free paclitaxel [118].

#### 11.2.5. Solid Lipid Nanoparticles (SLNs)

Solid lipid nanoparticles (SLNs) are tiny particles made up of solid lipids that are dispersed in an aqueous solution. They are smaller than liposomes and are known for their high stability and biocompatibility. SLNs have shown immense drug delivery potential for the treatment of non-small-cell lung cancer (NSCLC) due to their ability to improve drug efficacy and reduce toxicity. Studies have reported that different types of SLNs have been successful in enhancing the anti-tumor efficacy of drugs used to treat NSCLC. For instance, paclitaxel-loaded solid lipid nanoparticles (PTX-SLNs) were found to be effective in improving the anti-tumor efficacy of paclitaxel in lung cancer cells both in vitro and in a mouse model of NSCLC [119]. Similarly, another study reported the creation of an erlotinib-loaded solid lipid nanoparticle (ERL-SLNs)-based dry powder inhaler formulation. The synthesized ETB-SLNs exhibited high anticancer activity in lung cancer cells [120]. In a different study, findings confirmed that in a mouse inhalation model, the repeated inhalation exposure to SLN at concentrations lower than a 200-µg deposit dosage was safe [121]. Dry powder formulations using chitosan-derivative-coated solid lipid nanoparticles for inhalation have contributed to increased anti-cancer action in folate receptor-positive lung tumors [122]. According to a different study, epirubicin solid lipid nanoparticles (EPI-SLNs) could be used as an inhalable delivery strategy for lung cancer treatment [123].Finally, another study has reported that SLNs are a promising pulmonary delivery mechanism for enhancing the bioavailability of medicines that are weakly water soluble, such as naringeninin NSCLC [124].

#### 11.2.6. Nanostructured Lipid Carriers (NLCs)

Nanostructured lipid carriers (NLCs) are a drug delivery system similar to solid lipid nanoparticles (SLNs) but which have the added benefit of containing a mixture of solid and liquid lipids. Because of this, NLCs can encapsulate multiple drugs with different physicochemical properties, making them more versatile for accommodating a wider range of drugs. NLCs have been shown to improve the solubility and bioavailability of drugs and enhance their targeting ability. Docetaxel-loaded poloxamer-coated poly (lactic-co-glycolic acid) (PLGA) NPs were developed for the treatment of NSCLC via the pulmonary route. When compared to free DTX, these nanoparticles demonstrated greater cytotoxicity and regulated drug release, indicating improved treatment efficacy [125]. In vivo investigations demonstrated that the optimised formulation of nano-lipid carriers (NLCs) containing Paclitaxel (PTX) and Doxorubicin (DOX) exhibited improved retention and drug accumulation in the lungs with no evidence of tissue abnormalities and helped to reduce harmful effects in non-target tissues. Furthermore, both the cell line and in vivo results revealed that low-medication doses could be effective in achieving targeted therapeutic outcomes. These findings support the hypothesis that the pulmonary delivery of chemotherapeutics via an adequate inhalable nano-lipid carrier could be a promising chemotherapeutic method for lung cancer [126].

#### 11.2.7. Lipid-Polymer Hybrid Nanoparticles (LPHN)

Hybrid nanoparticles are a combination of lipids and polymers that have the potential to improve a drug’s loading and release properties. They can overcome biological barriers and increase therapeutic effectiveness, making them attractive platforms for drug delivery. These nanoparticles are designed to target non-small-cell lung cancer (NSCLC) by incorporating tumor-targeting ligands or antibodies onto their surfaces. This allows them to selectively bind to tumor cells and deliver drugs directly to the cancerous tissue.

Studies have shown that inhalable hybrid lipid protein nanoparticles nanocomposites that are dual-targeted can deliver genistein and all-trans retinoic acid for lung cancer therapy [127]. Aptamer-decorated lipid-polymer hybrid nanoparticles (ALPNPs) have also been found to have high drug loading capacities and good stability and are efficiently taken up by lung cancer cells in vitro. Co-delivery of docetaxel prodrug and cisplatin in the ALPNPs resulted in synergistic cytotoxicity against lung cancer cells compared to either drug alone. Another study reported that inhalation formulations based on silibinin-loaded chitosan-coated PLGA/PCL nanoparticles improved cytotoxicity and bioavailability in lung cancer patients [128].

#### 11.2.8. Polymeric Nanoparticle Delivery Systems

Polymeric nanoparticles are synthesized using biodegradable polymers such as poly (lactic-co-glycolic acid) (PLGA) and polyethylene glycol (PEG). These nanoparticles are capable of encapsulating drugs for targeted delivery, and they can release the drugs in a controlled and sustained manner. This feature of polymeric nanoparticles makes them an attractive option for drug delivery in the treatment of lung cancer, especially non-small-cell lung cancer (NSCLC).

According to a recent study on A549 cells, the afatinib-loaded PLGA nanoparticles dry powder inhaler had improved penetration and a stronger cytotoxic capability at a lower IC50 value than the plain afatinib dimaleate solution [129].

#### 11.2.9. Gold Nanoparticle Delivery Systems

Gold nanoparticles are tiny biocompatible particles that can be functionalized with targeting ligands in order to selectively target cancer cells. They have been used as a passive targeting strategy for non-small-cell lung cancer (NSCLC), as they can accumulate in tumors through the EPR effect and can be imaged using computed tomography (CT) scans [130]. Researchers have developed gold nanocages coated with hyaluronic acid (HA) for the targeted delivery of docetaxel to the NSCLC cells that overexpress CD44, a receptor for HA. These nanocages were shown to selectively accumulate in CD44-positive NSCLC cells and enhance the cytotoxicity of docetaxel. In addition, a multifunctional AuNP-based nanoplatform was developed for the targeted co-delivery of cisplatin and siRNA to NSCLC cells. According to the most recent research, inhalation is the best route for delivering gold nanoparticles to the lung, with high deposition efficiency, good distribution, tissue integrity preservation, and no evidence of caused inflammation [131].

#### 11.2.10. Use of pH-Sensitive Drug Delivery Systems

A recent study demonstrated that the pulmonary co-delivery of doxorubicin and siRNA via pH-Sensitive nanoparticles has an improved antitumor activity as compared to the single delivery method [132]. According to one study, nanoparticles of pH-responsive PEG-Doxorubicin conjugates were rapidly released at acidic pH. They spread easily in propellant-based metered-dose inhalers [133].

Overall, pH-sensitive drug delivery systems offer a promising approach for targeted drug delivery to tumors, including NSCLC. These systems can be tailored to respond to the unique properties of a tumor’s microenvironment, such as its acidic pH, and can deliver drugs with increased efficacy and reduced toxicity.

**Table 1 pharmaceutics-15-02777-t001:** Examples of current pulmonary-delivered medicines and repurposed medicines investigated for lung cancer.

API Type	Formulation Type	Delivery Method	Reference
Azacitidine	Solution	Aerosol, Nebulizer	[134]
Azacitidine	Solution/Dry powder	Aerosol, Nebulizer/Dry powder, nose-only inhalation	[135]
Bevacizumab	Dry powder	Aerosol, nasal inhalation	[136]
2-ME 2-methoxyestradiol	Nanocomposites and nanoaggregates	Intratrchial insufflation	[137]
5-Fluorouracil	Lipid coated nanoparticles	Inhalation Aerosol, Nebulizer	[138,139]
Epirubicin	Solid Lipid Nanoparticles (SLNs)	Inhalation Aerosol, Nebulizer	[123]
9-Nitrocampthotecin	Liposome	Inhalation Aerosol, Nebulizer	[140,141,142]
Afatinib & paclitaxel	Lipid-based nanocarriers	In vitro: Turbospin (single dose powder inhaler device). In vivo: Dry powder insufflator—Lipid-based nanocarriers	[129,143]
Camptothecin	Aerosolized liposomal Camptothecin	Inhalation Aerosol, Nebulizer	[144]
Carboplatin	Solution	Inhalation Aerosol, Nebulizer	[145]
Doxorubicin Celecoxib	Liposomes, EGF-modified gelatine nanoparticles, PLGA microparticles, drug conjugates	Inhalation, Aerosol, pMDI	[114,146,147,148,149,150,151]
	Lipid-based nanocarriers	Inexpose™ nebulizer Nanolipidcarriers	[152]
Celecoxib & Docetaxel	Solution	Inhalation Aerosol, MDI	[153,154]
Cisplatin	SLIT: Lipid vesicles	Inhalation Aerosol, nebulizer	[155]
Cisplatin loaded EGF-modified GP	Gelatin nanoparticle	Endotrachial installation	[156]
CpG & Poly I:C	Liposomal formulations	Intratracheal instillation	[157]
Curcumin	Nanocomposites and nanoaggregates	In vitro: Aerosol using cascade impactors	[158]
	Liposomal formulations	Intratracheal instillation	[159]
	Liposomal formulations	IntratrachialInsufflator	[117]
Curcuminoids	Lipid-based nanocarriers	Aerosol, Side stream jet nebulizer	[160]
Methotroxate	HFA-based Microparticles	Aerosol, Metered Dose Inhaler	[161]
Docetaxel	Liposomal formulations	Intratracheal administration	[116]
Docetaxel	Lipid-based nanoemulsion	OMRON MicroAIR nebulize	[162]
Docetaxel	Nanoparticles	DPI	[125]
Docetaxel and Curcumin	Nanoemulsion	In vitro: Aerosol, Anderson cascade impactor	[163]
Doxorubicin	Poly(butyl cyanoacrylate) nanoparticles	Aerosol, DPI	[164]
	Effervescent nanoparticles	Intratrachial insufflator	[165]
	Highly porus large PLGA microparticles	Inhalation	[28,133,144,166,167,168,169,170]
	Liposomal formulations	Intratracheal administration using microsprayer	[171]
	Solution (0.4 to 9.4 mg/m^2^)	Aerosol, Nebulizer	[172]
	Liposomal formulations	Aerosol, One-jet Collison nebulizer	[173]
Doxorubicin & ASO, or siRNA	LHRH receptor-targeted mesoporous silica nanoparticles	Inhalation	[114,132]
Doxorubicin and paclitaxel	Lipid-based nanocarriers	Collision nebulizer connected to nose-only exposure chamber	[115,126]
Epirubicin	Solid lipid nanoparticles	Aerosol, nebulizer	[123]
Erlotinib	Microparticles	Aerosol, DPI	[120]
Gefitinib	Lipid-based nanocarriers	Intratracheal installation	[174]
Cisplatin	EGF-modified Gelatin Nanoparticles (LNPs)	Aerosol, nebulizer	[27]
Gemcitabine & cisplatin	Niosomes	Aerosol, nebulizer	[175]
Gemcitabine	Solution	Aerosol, nebulizer	[176,177,178]
Retinoic acid and and genistein	Nanoparticles	DPI	[127]
Gemcitabine-HCl	Liposomal formulations	Intratracheal/Insufflator (Aerosol)	[179]
HC & 5-Amino levulinic acid	Cationic Liposomal nanoparticles	Endotrachial installation	[180]
Hyaluronan (HA)-cisplatin conjugates	Drug conjugates	Endotrachial installation	[181]
IL-2	Liposomes	Aerosol, nebulizer	[181]
Doxorubicin	Solution	Aerosol, nebulizer	[113]
Cyclosporin A and paclitaxel	Liposomes	Aerosol nebulizer	[182]
Doxorubicin	Self-assembled albumin nanoparticles	Aerosol, nebulizer	[113]
Anti-carbonic anhydrase IX (CA IX) antibody, conjugated to the surface of triptolide (TPL)	Liposomes	Aerosol	[118]
9-Bromo-noscapine	Lipid-based nanocarriers	Inhalation by indigenously developed apparatus	[183]
Myricetin	Nanoencapsulated Phospholipid Complex	In vitro: Aerosol using Aerolizer connected to Anderson Cascade Impigner	[184]
Silibinin	Nanoparticles	Inhalation, DPI	[128]
Nitro-camptothecin	Liposomes	Aerosol, nebulizer	[185,186,187]
Oridonin	DLPC liposome	Aerosol jet nebulizer	[188]
Paclitaxel	Liposomes	Aerosol, nebulizer	[168]
	Lung surfactant mimetic and pH-responsive lipid nanovesicles	Inhalation	[139]
	Chitosan-coated folate-PEG nanoparticles	Endotracheal administration, Micro Sprayer Aerosolizer IA-1C	[119]
	Lipid-based nanocarriers	DP insufflator	[189]
	Lipid-based nanocarriers	Aerosol, Collison nebulizer	[190]
	SLN- solid lipid nanoparticle	DPI- dry powder inhaler	[122]
Phospho-sulindac	Liposome	Aerosol, nebulizer	[191]
Quercetin	Lipid-based nanocarriers	OMRON MicroAIR nebulizer	[192,193,194]
Naringenin	SLN	Intratracheal instillation	[124]
Resveratrol	Nanocomposites and nanoaggregates	Inhalation	[195]
siRNA	DOTAP-modified PLGA nanoparticles	In vitro: aerosol generated by small scale powder disperser	[196]
Sorafenib Tosylate	Liposomal formulations	In vitro: DPI, Revolizer device	[197]
TAS-103	PLGA Nanocomposites and nanoaggregates	DPI, insufflator	[198]
Temozolomide	Liposomal formulations	Intratracheal administration using Microsprayer IA-1C system	[199]
	Nanocomposites and nanoaggregates	In vitro: DPI, Axahaler^TM^	[200]
	Liposomal formulations	Micro sprayer Aerosolizer Pulmonary Aerosol Kit for Mouse Model PAK-MSA	[201]
Amodiaquine	Nanoparticles	Inhalation, aerosol	[202]
Pioglitazone	Powder	Inhalation, aerosol	[203]
Telmisartan	Nanoparticles	Intratumoral distribution	[204]
Amodiaquine	Inhalable nanoparticulate system	Inhalation, nebulizer	[202]
Bexarotene (Targretin) & budesonide	Powders	Inhalation, aerosol, turbuhaler	[205,206,207]
Itraconazole	Dry powder for inhalation	In vitro: Cyclohaler™ Dry Powder Inhaler, twin stage impigner apparatus.	[208]
	Dry powder for inhalation	Inhalation Aerosol, DPI	[209]
	Solid dispersion	Inhalation Aerosol, DPI	[210]
Fisetin	Dry powder	Aerosol, DPI	[211]
Isotretinoin	Powder	Aerosol	[212]
Metformin	Sterosomes	Aerosol, nebulizer	[213]
Pirfenidone	Liposome	Microsprayer^®^ Aerosolizer Pulmonary Aerosol-Kit for Mouse	[214]

The pioneering work by Cory et al. in exploring the various delivery systems for 5-fluorouracil (5-FU) has significantly contributed to lung cancer therapy. Their investigation into liposomes, microspheres, and lipid-coated nanoparticles (LNPs) has provided crucial insights. In particular, the varying release times, with liposomes releasing 5-FU within 4–10 h and microspheres composed of poly-(lactide-co-glycolide) (PLGA) and poly-(lactide-co-caprolactone) (PLCL), showed slower release rates and demonstrated the potential for customizable delivery durations. This is crucial for tailoring treatments to specific patient needs [138].

Building on this context, another study by Cory et al. examined the inhalation delivery of 5-FU in LNPs using a hamster model. The research highlighted the distribution and concentration of 5-FU in different tissues post-inhalation, revealing initial high concentrations in the trachea, larynx, and esophagus, and comparatively lower levels in the lung. This distribution pattern underlines the feasibility of LNPs for lung cancer chemotherapy through inhalation [139].

Bart et al.’s study on aerosolized SLIT cisplatin in lung carcinoma patients further supports this potential. The treatment was well-received, showing minimal systemic toxicity and manageable side effects, which were predominantly gastrointestinal and respiratory. The pharmacokinetic analysis revealed low plasma platinum levels, which were indicative of effective local delivery, bolstering the case for aerosolized SLIT cisplatin as a viable lung cancer treatment [155].

Similarly, Zhang et al.’s investigation into liposomal gefitinib dry powder inhalers (LGDs) for non-small-cell lung cancer (NSCLC) treatment aligns with these findings. The LDGs’ rapid absorption from the lung, their higher concentration and retention as compared to oral administration, and their reduced inflammatory response and potential for lung injury, all point to the superiority of this method for NSCLC treatment [174].

In a parallel vein, Kim et al. focus on doxorubicin-loaded PLGA microparticles (Dox PLGA MPs), revealing their high encapsulation efficiency and effective aerosolization characteristics. After pulmonary administration, these MPs demonstrated prolonged lung retention and significant in vitro cytotoxicity against melanoma cells. The resulting smaller tumor sizes in a melanoma metastasis mouse model illustrate the potential of Dox PLGA MPs for long-term inhalation therapy in lung cancer [147].

In the area of in vivo activity, Koshkina and Kleinerman investigated the efficacy of aerosolized gemcitabine for treating osteosarcoma lung metastases. Aerosol GCB’s superiority over intraperitoneal administration in inhibiting lung metastases growth, coupled with its non-toxic profile, showcases its potential as an effective treatment modality [176]. The authors in other studies evaluated the aerosolized paclitaxel (PTX) in liposomal formulations, which similarly demonstrated a significant reduction in lung tumour burdens and enhanced survival rates in a murine model of renal carcinoma with pulmonary metastases [168]. Shepard et al. contributed to this growing body of research by developing a dry powder pulmonary formulation of bevacizumab, a monoclonal antibody used in NSCLC treatment. This formulation’s suitability for deep lung delivery, coupled with the maintenance of bevacizumab’s anti-VEGF bioactivity and its efficacy at lower doses than intravenous controls in a rat NSCLC model, underscores the potential of such innovative formulations [136].

The development of an inhalable, stable dry powder formulation of 5-azacytidine (5AZA), an epigenetic therapy drug by Kuehl et al., further exemplifies this trend. The formulation’s superior pharmacokinetics and its effectiveness in reducing the tumour burden in an orthotopic rat lung cancer model, demonstrates its ability to reprogram the cancer genome and marks a significant advancement in lung cancer treatment [135].

The study on phospho-sulindac, a derivative of sulindac with anticancer activity, addresses the challenges of metabolic stability that limit its use in systemic therapy. The aerosolized form of phospho-sulindac, developed to overcome first-pass metabolism, showed high levels of the intact drug in the lungs and significantly inhibited lung tumorigenesis. This method not only improved the survival of mice bearing orthotopic A549 xenografts but also minimized hydrolysis to less active metabolites [191].

Based on combination therapy, a study by Koshkina et al. examined the co-delivery of cyclosporine A paclitaxel via liposomal aerosol in a Renca-lung-metastases mouse model. The combination yielded favorable effects on tumor growth, although weight loss in the treated mice suggested potential toxicity in need of management. However, no significant toxicity was observed histopathologically [182].

In the clinical setting, Zarogoulidis et al. explored inhaled chemotherapy as an alternative for NSCLC treatment. Patients receiving a combination of intravenous and inhaled carboplatin showed a significant increase in survival rates, suggesting the efficacy of inhaled carboplatin as an alternative pulmonary drug delivery method [145].

## 12. Use of Repurposed Inhaled Anticancer Agents for Targeting NSCLC

Over the past few years, there has been a growing interest in repurposing inhaled anticancer agents for the treatment of non-small-cell lung cancer (NSCLC). Several inhaled anticancer agents have shown promise in targeting NSCLC, as indicated in Table 1. For instance, paclitaxel, a drug that stabilizes microtubules and inhibits cell division, has been found to be effective when inhaled using a nebulizer in the preclinical models of NSCLC [215]. In a phase II clinical trial, the combination of paclitaxel and carboplatin was linked with significant survival rates and moderate toxicity in fit, older patients with advanced NSCLC [216]. Another potential candidate for being repurposed as an anti-cancer therapy is amodiaquine, a drug that is commonly used to treat malaria. In pre-clinical studies, amodiaquine was shown to have anti-cancer properties, especially for NSCLC [202]. The drug has been found to inhibit the growth of cancer cells and induce cell death in vitro. Further studies are needed to evaluate its efficacy in treating NSCLC in vivo. Liposomal cisplatin is another inhaled anticancer agent that has shown promise for treating NSCLC. Cisplatin is a chemotherapy drug that can effectively kill cancer cells, but its use is limited due to its toxic side effects. Liposomal cisplatin is a reformulated version of cisplatin, which is encapsulated in a liposomal carrier to reduce its toxicity.

Several examples of repurposed drugs that are used to target NSCLC are listed in Table 1. These drugs have been approved for other indications and are being repurposed for NSCLC treatment based on their potential anticancer properties. Repurposing these drugs can provide a faster and more cost-effective approach to drug development, as these drugs have already undergone safety and efficacy testing for their original indications. The pulmonary route appears to be a viable strategy for reducing the significant systemic toxicity associated with chemotherapy. In phase I, Ib/IIa, and II clinical trials, inhaled chemotherapy was shown to be both practical and safe. Inhalation permits strong therapeutic doses to be delivered directly to lung tumours without prior distribution in the body. As a result, any severe systemic toxicities are decreased [217]. A recent review showed that as a local treatment, inhaled cytotoxic chemotherapy is a potential therapy for bridging the gap between local and broad systemic treatments. It could concentrate the dose in the lungs and gradually disperse it into the blood and lymph systems, which are the primary pathways for cancer invasion [218].

With the introduction of repurposed and targeted medications that offer greater therapeutic alternatives, non-small-cell lung cancer (NSCLC) treatment has witnessed tremendous breakthroughs. It can be seen from previous research that several medications have been tested for their anticancer activities such as, but not limited to, amodiaquine inhalable nanoparticles, which were successfully formulated and optimised by employing a systematic strategy of experiment design via a scalable high-pressure homogenization (HPH) approach [202]. Another study claimed that pioglitazone aerosol could be a beneficial drug for regional lung cancer prevention, and may even be used with other medicines to create a combination chemoprevention (e.g., retinoids, biguanides such as metformin, etc.) [203]. Since 1968, the inhalation of cytotoxic chemotherapy has been the focus of various research studies, although only a few clinical trials have been conducted so far [142,145,155,172,178,219,220,221] (Table 2). These trials have revealed that inhalation can result in lower or minimal systemic side effects, offering an improved pharmacokinetic profile. However, a major challenge in these trials is the exclusive use of nebulizers, including jet, ultrasonic wave, and vibrating mesh types, to administer the chemotherapy. Typically delivered via intravenous perfusion, these nebulizers aim to convert the chemotherapy solution into fine droplets smaller than 5 μm for direct lung inhalation. However, a significant portion of the aerosol is lost during the nebulization, affecting the effectiveness of this delivery method [222,223,224].

Cytotoxic chemotherapy comprises hazardous drugs, necessitating extensive protective measures for healthcare personnel to minimize the exposure risks [225]. Achieving effective lung deposition might require large drug doses administered over extended periods through nebulization, which raises health concerns [155]. Additionally, the infusion of certain cytotoxic drugs, which make up about 10–30% of lung cancer chemotherapies, can cause pulmonary toxicities, leading to clinician hesitancy in using this treatment approach [226].

Technological and pharmaceutical advancements offer new possibilities for inhaled cytotoxic chemotherapy, contingent upon addressing key clinical and technological factors. The administration time for aerosolized drugs varies based on dose and concentration, and the efficiency of nebulizers in depositing the drug in the lungs also differs. For example, a jet-nebulizer delivered only 10–15% of a radiolabelled CIS dose to the lungs, whereas a vibrating mesh nebulizer achieved 43% lung deposition with a radiolabelled gemcitabine solution. This low efficiency imposes a critical limitation on delivering effective treatment [178,219]. Additionally, the pressure generated by the nebulizer, which influences the rate of nebulization (up to 0.3 mL/min), also impacts how quickly and efficiently the drug is delivered to the patient’s lungs and plays a significant role in the overall success of the therapy [155].

**Table 2 pharmaceutics-15-02777-t002:** Clinical studies investigating the safety and efficacy of inhaled chemotherapy for NSCLC.

Drug	Phase	Device, Formulation	Patient Status (*n*)	Deposition in the Lung	Local Dose Limiting Toxicity	Severe Systemic Toxicity	Disease Response (*n*)	Reference
Azacitidine	Phase 1/2	Aerosol, Nebulizer/Dry powder, nose-only inhalation	Local and metastatic lung cancer	Direct deposition into the bronchi and lung	No pulmonary toxicity	Pale skin, shortness of breath, fast heartbeat, chest pain, cough, unusual bruising or bleeding	(29%) of patients had stable disease with one partial, and one complete response	[134,135]
5-FU	Pilot	Wave nebulizer, iv solution	Lung cancer [222], lung metastasis [223]	5–15 times more concentrated in tumor compared to lung tissues	None	None	Complete response [142], Partial response [219], progressive disease [219]	[172]
9-nitro-camptothecin	I	Jet nebulizer, liposome dispersion	Lung cancer and lung metastases [70]	4–10 time more concentrated in the bronchoalveolar lavage compared to serum	Grade 2: cough, bronchial irritation Grade 3: Chemical pharyngitis	Grade 2: nausea, vomiting, anaemia, neutropenia	Partial response [155], stable disease [155]	[142]
Cisplatin	I	Jet nebulizer, liposome dispersion	Advanced NSCLC [61], SCLC [172]	10–15% (radiolabelled solution)	Grade 3: Bronchitis, dyspnea, decreased lung function	Grade 3: Fatigue Grade 4: thrombosis	Stable disease [225], progressive disease [219]	[155]
	Ib/IIa		Osteosarcoma with lung metastases [64]	N/A	Grade 2: Hoarseness	Grade 3: Nausea, vomiting	Partial response [172], stable disease [178], progressive disease [145]	[219]
Doxorubicin	I	Jet nebulizer, solution	Lung metastases: [221] sarcoma [64], Osteosarcoma [219], NSCLC [61], colorectal [221], thyroid [155], Miscellaneous [221]	Correct deposition in the lung (radiolabelled solution)	Grade 2: Cough, wheezing, dyspnea Grade 3: Hypoxia Grade 4: Respiratory distress, dyspnea	No Grade 3/4 toxicity observed	Partial response [172], stable disease [145], progressive disease [142]	[221]
Doxorubicin (inhaled) + cisplatin (iv) and docetaxel (iv)	I/II	Jet nebulizer, solution	Advanced NSCLC [227]	N/A	Grade 3–4: Cough, decrease in pulmonary function test	Grade 3–4: Constipation, hyponatremia, neutropenia	No significant improve in survival, most patients had stable disease	[221]
Gemcitabine	I	Mesh nebulizer, iv solution	NSCLC [224]	42 ± 16% (homogenous deposition)	Grade 2–3: Cough Grade 4: bronchospasm	Grade 3: Fatigue, vomiting	Minor response [172], stable disease [219], progressive disease [219]	[178]
Carboplatin	I/II	Jet nebulizer, solution	NSCLC [145]	Deposition in lung parenchyma (radiolabelled solution)	Grade 2–3 Cough Grade 3: Dyspnea, hoarseness	Grade 3: Fatigue, alopecia, rash, anorexia, anemia, neutropenia, pharyngitis, mucositis Grade 4: anorexia, neutropenia	No significant improve in survival, most patients had progressive disease	[145]

## 13. Conclusions

This review provides an in-depth analysis of the different drug delivery systems for the lungs that have been developed to address non-small cell lung cancer (NSCLC). It begins by explaining the anatomy and physiology of the respiratory system, highlighting the essential roles of both the upper and lower tracts in facilitating efficient drug delivery. This article also discusses various inhalation devices ranging from pressurized petered-dose inhalers, dry powder inhalers, nebulizers, and soft mist inhalers, underscoring the range of options available and their unique advantages in delivering therapeutic agents to specific lung regions. The success of each treatment is determined by several factors such as the inhalation technique, the disease state, and the adherence to prescribed regimens. The article also explores the use of carrier-free nanodrugs, which offer a green and straightforward synthesis technique that holds promise for longer drug retention and reduced side effects. Targeting strategies for NSCLC have also advanced significantly in recent years, including active targeting techniques that leverage monoclonal antibodies or small molecule inhibitors that promise enhanced specificity. Passive targeting mechanisms, such as gold nanoparticles, have potential for tumor accumulation and imaging capabilities and also hold promise for NSCLC treatment. Collectively, these diverse innovations, when harmoniously integrated, have the potential to dramatically transform the therapeutic landscape for NSCLC, heralding a future with more effective, tailored, and patient-centric treatments. 

## Figures and Tables

**Figure 1 pharmaceutics-15-02777-f001:**
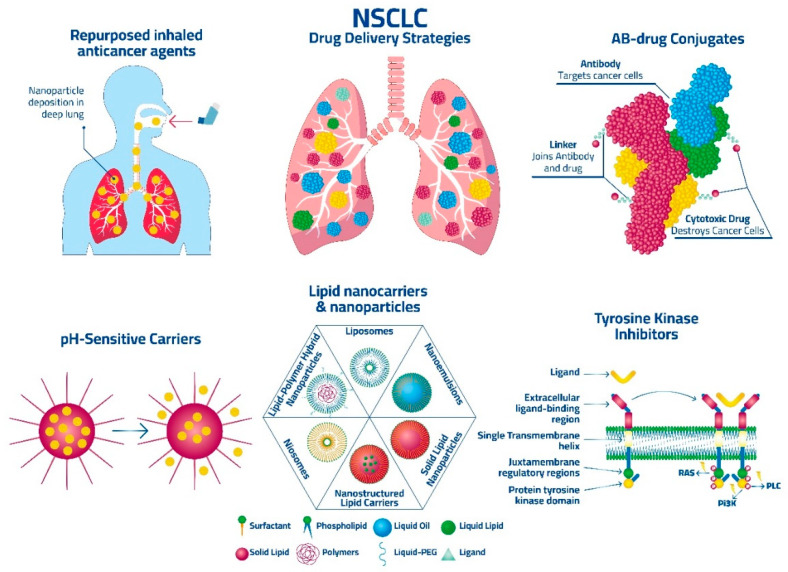
Examples of current drug delivery strategies for the treatment of NSCLC.

## Data Availability

Not applicable.

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
