# Peer review of "Inhaled Medicines for Targeting Non-Small Cell Lung Cancer"

_pharmaceutics, 2023, doi:10.3390/pharmaceutics15122777_

Round 1

Reviewer 1 Report

Comments and Suggestions for Authors

The manuscript entitled "Inhaled Medicines for Targeting Non-Small Cell Lung Cancer" is a detailed review of the up-to-date methods for delivering medications directly to the lungs.
It is a well-written review that introduces the different types of lung cancer and then focuses on a particular one, the Non-Small Cell Lung cancer. The authors describe a set of therapies currently used to treat this kind of tumor and the strategies and caveats of these inhaled treatments.
All in all, I think it is a nice review, that is well written and supported by an extensive collection of bibliographical references to underpin each section.
That said, I would like to see a table compiling (at least) the top ten inhaled anti-cancer therapies (currently used) and their respective 5-year relative survival rate, to improve the reader's understanding of the manuscript and present an applied medicine perspective to the review.

Comments on the Quality of English Language

The quality of english seems correct.

Author Response

Response to reviewer’s comments

We would like to express our appreciation to the reviewer who generously dedicated their valuable time to provide us with constructive comments and insightful recommendations. Their feedback has immensely improved the quality of the manuscript. 

The manuscript entitled "Inhaled Medicines for Targeting Non-Small Cell Lung Cancer" is a detailed review of the up-to-date methods for delivering medications directly to the lungs.
It is a well-written review that introduces the different types of lung cancer and then focuses on a particular one, the Non-Small Cell Lung cancer. The authors describe a set of therapies currently used to treat this kind of tumor and the strategies and caveats of these inhaled treatments.
All in all, I think it is a nice review, that is well written and supported by an extensive collection of bibliographical references to underpin each section.
That said, I would like to see a table compiling (at least) the top ten inhaled anti-cancer therapies (currently used) and their respective 5-year relative survival rate, to improve the reader's understanding of the manuscript and present an applied medicine perspective to the review.

We would like to express our gratitude for your insightful suggestion to incorporate a table that compiles the top ten inhaled anti-cancer therapies and their respective 5-year relative survival rates. Your recommendation is truly appreciated, as we recognize the value that such a table would add to the manuscript by providing an applied medicine perspective.

However, it is imperative to note that, at present, no inhaled chemotherapy treatments have yet received approval for human application. The field of inhaled chemotherapy, particularly for the treatment of Non-Small Cell Lung Cancer (NSCLC), is still in the early phases of clinical trials (phase I/II). As a result, there is a dearth of comprehensive data on the long-term survival rates, such as the 5-year relative survival rates, for these treatments.

In our manuscript, we have provided a detailed account of the current inhaled chemotherapy treatments that are currently undergoing clinical trials. This section provides an overview of these therapies and delves into the reasons behind their limited progress in clinical trials, such as issues related to drug delivery efficiency, aerosolization challenges, and patient safety concerns. We are confident that this information will offer readers a clear understanding of the present state of inhaled chemotherapy in oncological treatment, along with the hurdles that need to be overcome for their successful implementation.

Reviewer 2 Report

Comments and Suggestions for Authors

This paper is very nicely written and the grammar is excellent.

The major feedback I will give is that the background information is often too general and not directly relevant to the topic at hand: inhaled therapies for NSCLC. This makes the article too long given the subject matter, and redundant for a reader in the field.

Here are suggestions for sections which should be eliminated or shortened:
1. The Introduction section which focuses on basic lung anatomy is not needed at all.

2. The SCLC treatment details in this section can also be eliminated, as they are out of scope.

4. The section on pulmonary surfactants should be shortened, and more focus given to their impact on performance of inhaled cancer therapies

5. The MCC section is nice, but again, lacks context of how it impacts inhaled cancert therapies specifically

6. The section on devices should be shortened significantly, as it is covered extensively in other reviews, which can simply be cited.

7. This section on drug deposition is another topic which is covered extensively in other reviews and its relevance to this particular topic needs to be better emphasized. 

8. Same for section 8. The challenges outlined here are for all of inhalation delivery. Please either remove this section or recontextualize it within the lens of cancer treatment.

On page 13, we finally get to the topic at hand! Section 9 could easily be section 2 or 3 once the other background information is trimmed.

The first paragraph of section 9, particularly the second half, seems under-referenced. "They claim an impressive drug-carrying capacity and exhibit reduced side effects. Their engineered structures enhance accumulation within tumors. Furthermore, the synthesis process of these carrier-free nano drugs is both simple and environmentally friendly, often avoiding the use of harmful reagents like organic solvents. These nano drugs also demonstrate prolonged retention in the bloodstream compared to standard anti-cancer drugs." None of these bold claims have references.

Reading further into section 9...its not clear that the work on carrier-free nano drugs was even done for inhalation purposes? If that's the case, this section seems pretty irrelevant.

In section 10, this is again relevant to NSCLC, and a nice section, but we're still not talking about studies which used lung delivery. It is necessary to focus the review on the topic that your title states.

Section 10.2.1 The industry-accepted terminology would be "amorphous solid dispersions" not just solid dispersions. 
Section 10.2.2 only has one reference for the entire field of nanoparticle delivery. This makes no sense.

Table 1 has lots of good information, but could benefit from more columns to categorize the API type, formulation type, and delivery method, rather than listing them all in the central column.

First paragraph of section 11, please state what kind of inhalation device was used for these studies, was it a nebulizer or something else?

Just as the discussion of the actual topic of this paper begins, a table of repurposed drugs is shown and then conclusions. There is barely any detailed discussion of the papers cited which actually delivered cancer drugs to the lungs by inhalation. This is not useful.

Author Response

Response to reviewer’s comments

We would like to express our appreciation to the reviewer who generously dedicated their valuable time to provide us with constructive comments and insightful recommendations. Their feedback has immensely improved the quality of the manuscript.

Comments and Suggestions for Authors

This paper is very nicely written and the grammar is excellent.

The major feedback I will give is that the background information is often too general and not directly relevant to the topic at hand: inhaled therapies for NSCLC. This makes the article too long given the subject matter, and redundant for a reader in the field.

Thank you very much for this reviewer’s comments. 

Here are suggestions for sections which should be eliminated or shortened:

  1. The Introduction section which focuses on basic lung anatomy is not needed at all.

The basic lung anatomy data has been eliminated

  1. The SCLC treatment details in this section can also be eliminated, as they are out of scope.

All SCLC data and relevant treatment has been eliminated

  1. The section on pulmonary surfactants should be shortened, and more focus given to their impact on performance of inhaled cancer therapies

The pulmonary surfactants section has been shortened with focus only on their impact on performance of inhaled cancer therapies.

  1. The MCC section is nice, but again, lacks context of how it impacts inhaled cancer therapies specifically

A paragraph about the impact of MCC has been added at the end of section 5.Mucociliary clearance and mucoadhession, getting the balance right

  1. The section on devices should be shortened significantly, as it is covered extensively in other reviews, which can simply be cited.

Section 6 has been summarized efficiently eliminating all the details covered in previous reviews.

  1. This section on drug deposition is another topic which is covered extensively in other reviews and its relevance to this particular topic needs to be better emphasized. 

Section 7 Assessment of drug deposition, current methods and accuracy for assessment of deposition of anticancer agents, has been shortened and a paragraph focusing on its relevance to our topic has been added at the end of this section.

  1. Same for section 8. The challenges outlined here are for all of inhalation delivery. Please either remove this section or recontextualize it within the lens of cancer treatment.

Section 8 has been redefined within the lens of cancer treatment.

On page 13, we finally get to the topic at hand! Section 9 could easily be section 2 or 3 once the other background information is trimmed.

All previous section have been trimmed as the reviewer’s advice.

The first paragraph of section 9, particularly the second half, seems under-referenced. "They claim an impressive drug-carrying capacity and exhibit reduced side effects. Their engineered structures enhance accumulation within tumors. Furthermore, the synthesis process of these carrier-free nano drugs is both simple and environmentally friendly, often avoiding the use of harmful reagents like organic solvents. These nano drugs also demonstrate prolonged retention in the bloodstream compared to standard anti-cancer drugs." None of these bold claims have references.

This paragraph has been cited with the respective references.

Reading further into section 9...its not clear that the work on carrier-free nano drugs was even done for inhalation purposes? If that's the case, this section seems pretty irrelevant.

A paragraph stating “Using advanced organic solution and spray drying particle engineering design, a research study has successfully created and manufactured lactose-free dry powder formulations including the two medicines fluticasone propionate and salmeterol xinafoate FP/SX, molecularly combined with mannitol as an excipient. Furthermore, the resultant spray-dried powders possessed vital characteristics for respiratory medication administration” has been added to this section.

In section 10, this is again relevant to NSCLC, and a nice section, but we're still not talking about studies which used lung delivery. It is necessary to focus the review on the topic that your title states.

In section 10 Strategies for drug targeting of NSCLC, as subsection 10.1.4   Personalized Inhaled Medicine Approaches has been added.

Section 10.2.1 The industry-accepted terminology would be "amorphous solid dispersions" not just solid dispersions. 

The word “amorphous” has been added to this section as the reviewer’s advice.

Section 10.2.2 only has one reference for the entire field of nanoparticle delivery. This makes no sense.

Three additional references have been added to section 10.2.2 Use of Nanoparticle Delivery Systems

Table 1 has lots of good information, but could benefit from more columns to categorize the API type, formulation type, and delivery method, rather than listing them all in the central column.

Table 1 has been restructured as per the reviewer’s advice.

First paragraph of section 11, please state what kind of inhalation device was used for these studies, was it a nebulizer or something else?

This type of the inhalation device has been clarified in this section.

Just as the discussion of the actual topic of this paper begins, a table of repurposed drugs is shown and then conclusions. There is barely any detailed discussion of the papers cited which actually delivered cancer drugs to the lungs by inhalation. This is not useful.

Two paragraphs have been added before the conclusion to discuss the topic as well as the use of repurposed drugs.

Reviewer 3 Report

Comments and Suggestions for Authors

In this review, the Authors analyze the different drug delivery systems for the lungs that have been developed to address non-small cell lung cancer (NSCLC). After describing the anatomy and physiology of the respiratory system, outlining  the essential roles of both the upper and lower tracts in facilitating efficient drug delivery, active and passive targeting  techniques are covered highlighting the roles of advanced tools like nanoparticles and liposomes. Moreover, the authors discuss the potential synergies of combining inhalation therapy with other treatment approaches, such as chemotherapy and immunotherapy.

The topic of the review is relevant and covers an attractive area of interest.

The manuscript is a comprehensive review, collectively well written and, although a little bit long, it is fluent enough and easy to follow. The 2 Tables and figure seem nice and clear.

Comments

Although the bibliography contains a large number of references (300), it is not completely adequate and updated, and it should be shortened and improved. This is an important point, since the bibliography represents the core of a review.

A number of references could be avoided and/or replaced with more adequate and recent ones.

Just as an example, the references 5-9 (Latimer, K.M.; Mott, T.F. Lung cancer: diagnosis, treatment principles, and screening. Am Fam Physician 2015; Koinis, F.; Kotsakis, A.; Georgoulias, V. Small cell lung cancer (SCLC): no treatment advances in recent years. Translational lung cancer research 2016; Früh, M.; De Ruysscher, D.; Popat, S.; Crinò, L.; Peters, S.; Felip, E. Small-cell lung cancer (SCLC): ESMO Clinical Practice Guidelines for diagnosis, treatment and follow-up. Annals of oncology 2013; Waqar, S.N.; Morgensztern, D. Treatment advances in small cell lung cancer (SCLC). Pharmacology & therapeutics 2017; Tsoukalas, N.; Aravantinou-Fatorou, E.; Baxevanos, P.; Tolia, M.; Tsapakidis, K.; Galanopoulos, M.; Liontos, M.; Kyrgias, G. Advanced small cell lung cancer (SCLC): new challenges and new expectations. Annals of translational medicine 2018) could be replaced with more recent ones, such as the following:

Thai, A.A.; Solomon, B.J.; Sequist, L.V.; Gainor, J.F.; Heist, R.S. Lung cancer. Lancet 2021; Rocco D, Sapio L, Della Gravara L, Naviglio S, Gridelli C. Treatment of Advanced Non-Small Cell Lung Cancer with RET Fusions: Reality and Hopes. Int J Mol Sci. 2023. Rocco D, Della Gravara L, Ragone A, Sapio L, Naviglio S, Gridelli C. Prognostic Factors in Advanced Non-Small Cell Lung Cancer Patients Treated with Immunotherapy. Cancers 2023.

In addition, concerning the relevant issue on the potential use of naturally occurring molecules in NSCLC therapy, recent findings describing antitumor effects of the natural cAMP elevating agent Forskolin in NSCLC cells  have been provided (Salzillo A et al, Eur J Cell Biol, 2023).

Notably, forskolin-loaded delivery systems have been proposed for clinical purposes (NLCs as a potential carrier system for transdermal delivery of forskolin. Lasoń E, Sikora E, Miastkowska M, Escribano E, Garcia-Celma MJ, Solans C, Llinas M, Ogonowski J. Acta Biochim Pol. 2018; A forskolin-loaded nanodelivery system prevents noise-induced hearing loss. An X, Wang R, Chen E, Yang Y, Fan B, Li Y, Han B, Li Q, Liu Z, Han Y, Chen J, Zha D. J Control Release. 2022).

At this regard, a brief comment in the text could be reported, along with the above related references, to support the potential therapeutic role of drug delivery systems containing natural occurring molecules in NSCLC.

Comments on the Quality of English Language

I noted some misreadings throughout the text.

Author Response

Response to reviewer’s comments

We would like to express our appreciation to the reviewer who generously dedicated their valuable time to provide us with constructive comments and insightful recommendations. Their feedback has immensely improved the quality of the manuscript.

Comments and Suggestions for Authors

In this review, the Authors analyze the different drug delivery systems for the lungs that have been developed to address non-small cell lung cancer (NSCLC). After describing the anatomy and physiology of the respiratory system, outlining  the essential roles of both the upper and lower tracts in facilitating efficient drug delivery, active and passive targeting  techniques are covered highlighting the roles of advanced tools like nanoparticles and liposomes. Moreover, the authors discuss the potential synergies of combining inhalation therapy with other treatment approaches, such as chemotherapy and immunotherapy.

The topic of the review is relevant and covers an attractive area of interest.

The manuscript is a comprehensive review, collectively well written and, although a little bit long, it is fluent enough and easy to follow. The 2 Tables and figure seem nice and clear.

Comments

Although the bibliography contains a large number of references (300), it is not completely adequate and updated, and it should be shortened and improved. This is an important point, since the bibliography represents the core of a review.

A number of references could be avoided and/or replaced with more adequate and recent ones.

Just as an example, the references 5-9 (Latimer, K.M.; Mott, T.F. Lung cancer: diagnosis, treatment principles, and screening. Am Fam Physician 2015; Koinis, F.; Kotsakis, A.; Georgoulias, V. Small cell lung cancer (SCLC): no treatment advances in recent years. Translational lung cancer research 2016; Früh, M.; De Ruysscher, D.; Popat, S.; Crinò, L.; Peters, S.; Felip, E. Small-cell lung cancer (SCLC): ESMO Clinical Practice Guidelines for diagnosis, treatment and follow-up. Annals of oncology 2013; Waqar, S.N.; Morgensztern, D. Treatment advances in small cell lung cancer (SCLC). Pharmacology & therapeutics 2017; Tsoukalas, N.; Aravantinou-Fatorou, E.; Baxevanos, P.; Tolia, M.; Tsapakidis, K.; Galanopoulos, M.; Liontos, M.; Kyrgias, G. Advanced small cell lung cancer (SCLC): new challenges and new expectations. Annals of translational medicine 2018) could be replaced with more recent ones, such as the following:

Thai, A.A.; Solomon, B.J.; Sequist, L.V.; Gainor, J.F.; Heist, R.S. Lung cancer. Lancet 2021; Rocco D, Sapio L, Della Gravara L, Naviglio S, Gridelli C. Treatment of Advanced Non-Small Cell Lung Cancer with RET Fusions: Reality and Hopes. Int J Mol Sci. 2023.

 Rocco D, Della Gravara L, Ragone A, Sapio L, Naviglio S, Gridelli C. Prognostic Factors in Advanced Non-Small Cell Lung Cancer Patients Treated with Immunotherapy. Cancers 2023.

Numerous sections of the document have been abbreviated and enhanced, leading to the elimination of references 5 through 9. These references have been replaced with more current ones as recommended.

In addition, concerning the relevant issue on the potential use of naturally occurring molecules in NSCLC therapy, recent findings describing antitumor effects of the natural cAMP elevating agent Forskolin in NSCLC cells  have been provided (Salzillo A et al, Eur J Cell Biol, 2023).

Notably, forskolin-loaded delivery systems have been proposed for clinical purposes (NLCs as a potential carrier system for transdermal delivery of forskolin. Lasoń E, Sikora E, Miastkowska M, Escribano E, Garcia-Celma MJ, Solans C, Llinas M, Ogonowski J. Acta Biochim Pol. 2018; A forskolin-loaded nanodelivery system prevents noise-induced hearing loss. An X, Wang R, Chen E, Yang Y, Fan B, Li Y, Han B, Li Q, Liu Z, Han Y, Chen J, Zha D. J Control Release. 2022).

At this regard, a brief comment in the text could be reported, along with the above related references, to support the potential therapeutic role of drug delivery systems containing natural occurring molecules in NSCLC.

A paragraph has been added at the end of section 2. Lung cancer, aetiology, and current practice supporting the potential therapeutic role of drug delivery systems containing natural occurring molecules in NSCLC and the two suggested references have been cited.

Reviewer 4 Report

Comments and Suggestions for Authors

In the manuscript authors are reviewing recent data on new routes of anticancer drugs delivery directly to the lungs. They focus on application of inhalation technology for treatment of non-small cell lung cancer.  The topic is very relevant since new delivery methods of the clinically approved cytostatics may potentiate the therapeutic and reduce unwanted side effects. From this point of view any creative approach that would enhance our knowledge is worth to be published.

Authors started with brief description of the upper and lower respiratory tract and discussed factors that fascilitate adhesion and traficing of the adsorbed drug molecules in the lung/cancer tissue. The available inhalation devices and specifity of their application is followed by the adequate drug formulation, including recent carrier-free technology and drug targeting strategies.

The most valuable part is the content of Table 1 and 2, summarizing literature examples of the common anticancer drugs along with formulation and methods of delivery (Table 1) and examples of repurposed drugs that might be used for treatment of lung cancer(s) (Table 2).

This review is very comprehensive and up to date. However it would be interesting to obtain information about performance of such formulation and delivery techniques in clinical trials and their level. For example added in Table 1 and 2). Not much information is given about the influence of size, shape and charge of these inhaled particles on their location in the respiratory tract along with their pharmacodynamics and pharmacokinetics.

Another important issue that is not discussed is local and systemic toxicity of the inhalled solid particles.

Summing up, although the article is well written and interesting, some more critical information should be appended to upgrade the quality of the paper. Major revision.

Comments on the Quality of English Language

Text in general is well edited. Technical Editor might change text under Figures from italics to normal.

Author Response

Response to reviewer’s comments

We would like to express our appreciation to the reviewer who generously dedicated their valuable time to provide us with constructive comments and insightful recommendations. Their feedback has immensely improved the quality of the manuscript.

Comments and Suggestions for Authors

In the manuscript authors are reviewing recent data on new routes of anticancer drugs delivery directly to the lungs. They focus on application of inhalation technology for treatment of non-small cell lung cancer.  The topic is very relevant since new delivery methods of the clinically approved cytostatics may potentiate the therapeutic and reduce unwanted side effects. From this point of view any creative approach that would enhance our knowledge is worth to be published.

Authors started with brief description of the upper and lower respiratory tract and discussed factors that fascilitate adhesion and traficing of the adsorbed drug molecules in the lung/cancer tissue. The available inhalation devices and specifity of their application is followed by the adequate drug formulation, including recent carrier-free technology and drug targeting strategies.

The most valuable part is the content of Table 1 and 2, summarizing literature examples of the common anticancer drugs along with formulation and methods of delivery (Table 1) and examples of repurposed drugs that might be used for treatment of lung cancer(s) (Table 2).

This review is very comprehensive and up to date. However it would be interesting to obtain information about performance of such formulation and delivery techniques in clinical trials and their level.

A new table "Table 3" has been created for this purpose.

For example added in Table 1 and 2). *Not much information is given about the influence of size, shape and charge of these inhaled particles on their location in the respiratory tract along with their pharmacodynamics and pharmacokinetics.

A new section has been added in section 8, discussing the impact of particle size and shape on lung deposition, and the assessment methods for anticancer drugs.

*Another important issue that is not discussed is local and systemic toxicity of the inhalled solid particles.

Local and systemic toxicity of the inhaled anticancer agents have been covered in the newly implemented Table 3.

Summing up, although the article is well written and interesting, some more critical information should be appended to upgrade the quality of the paper. Major revision.

Thank you for taking the time to review our manuscript. We appreciate the thorough feedback and constructive comments you provided. We carefully considered your suggestions for major revisions and addressed each of the concerns raised.

Comments on the Quality of English Language

Text in general is well edited. Technical Editor might change text under Figures from italics to normal.

The text under the figure has been changed to normal font.

Round 2

Reviewer 2 Report

Comments and Suggestions for Authors

Thank you to the authors for their revisions, which have moved in the right direction. Substantially shortening the background has made the paper more accessible to the reader.

There is still a major issue with the content of this review article. Sections 10 and 11 at first read seem to be discussing examples of therapeutic approaches for inhaled treatment of NSCLC, but nearly all of the references cited are for systemic administration. There needs to be a tighter focus on what has been done for lung delivery specifically. For example, in section 11.2.5 "Solid lipid nanoparticles," reference 123 is about inhaled delivery, while 124-126 are not inhaled delivery. This is confusing to the reader and misleading. The same appears to be true for many of these sections.

Section 12, where is the reference to the paclitaxel Phase 2 trial?

Table 1 is the heart of this review, and very interesting. However, there is barely any discussion of the references in table 1, when this should make up the majority of the paper. I suggest that the reviewers reconsider sections 10 and 11, as they are not related to inhalation delivery, and replace them instead with detailed discussion of the most interesting papers in Table 1. It would be reasonable to spend multiple pages on discussion of the content of this important table.   

Additionally regarding Table 1, Please be more specific with formulation type and delivery method. "Inhalation" is not helpful enough. In all cases, it would be helpful to know if it was a dry powder formulation (carrier based or carrier-free), a liquid formulation (solution, emulsion or suspension), and what kind of devices were used, if any (intratracheal insufflator, aerosol generator, nebulizer, etc.).

Missing from Table 1 are recent studies on inhaled azacitidine (Kuehl et al, Cheng et al), and inhaled bevacizumab (Shepard et al).

The idea of highlighting which drugs are repurposed approved drugs in Table 2 is nice. However, many of the drugs in table 1 are also examples of repurposed drugs, they just aren't included in table 2. Perhaps you could combine the tables and have a column highlighting which of the actives are already approved, both as cancer drugs or for other indications.

The table highlighting clinical studies is nice. Again, an inhaled azacitidine study is missing (Cheng et al 2021, Lung Cancer).

Author Response

We would like to express our gratitude for  the reviewer's consideration in reviewing our submitted review. The reviewer's expertise in the field is highly valued, and we would like to confirm considering their constructive comments as follows:

Thank you to the authors for their revisions, which have moved in the right direction. Substantially shortening the background has made the paper more accessible to the reader.

  • Comment: There is still a major issue with the content of this review article. Sections 10 and 11 at first read seem to be discussing examples of therapeutic approaches for inhaled treatment of NSCLC, but nearly all of the references cited are for systemic administration. There needs to be a tighter focus on what has been done for lung delivery specifically. For example, in section 11.2.5 "Solid lipid nanoparticles," reference 123 is about inhaled delivery, while 124-126 are not inhaled delivery. This is confusing to the reader and misleading. The same appears to be true for many of these sections.
  • - Response: All references from sections 11.2.1 to 11.2.9 have been updated to include inhalation therapies for lung delivery instead of systemic administration.
  • Comment: Section 12, where is the reference to the paclitaxel Phase 2 trial?
  • Response: The reference to the phase 2 trial of paclitaxel has been added.

  • Comment: Table 1 is the heart of this review, and very interesting. However, there is barely any discussion of the references in table 1, when this should make up the majority of the paper. I suggest that the reviewers reconsider sections 10 and 11, as they are not related to inhalation delivery, and replace them instead with detailed discussion of the most interesting papers in Table 1. It would be reasonable to spend multiple pages on discussion of the content of this important table.   
  • Response: Thank you for the suggestion provided by the reviewer. However, we would like to keep section 10, as delivering anticancer drugs to the lungs often requires higher doses, and carrier-free technologies are necessary to achieve such high doses. Similarly, in section 11, we discussed strategies for drug targeting of NSCLC, which can be achieved through the inhaled route as well. We have added Table 1 to provide a summary of current inhaled medicines for the treatment of lung cancer, enabling interested readers to quickly find relevant studies if needed.

  • Comment: Additionally, regarding Table 1, Please be more specific with formulation type and delivery method. "Inhalation" is not helpful enough. In all cases, it would be helpful to know if it was a dry powder formulation (carrier based or carrier-free), a liquid formulation (solution, emulsion or suspension), and what kind of devices were used, if any (intratracheal insufflator, aerosol generator, nebulizer, etc.).
  • Response: Table 1 has been restructured as per the reviewer’s advice.

  • Comment: Missing from Table 1 are recent studies on inhaled azacitidine (Kuehl et al, Cheng et al), and inhaled bevacizumab (Shepard et al).
  • Response: Both drugs Azacitidine and Bevacizumab have been added to table 1.

  • Comment: The idea of highlighting which drugs are repurposed approved drugs in Table 2 is nice. However, many of the drugs in table 1 are also examples of repurposed drugs, they just aren't included in table 2. Perhaps you could combine the tables and have a column highlighting which of the actives are already approved, both as cancer drugs or for other indications.
  • Response: All repurposed drugs have been removed from Table 1 and added to Table 2 in order to organize the data in the tables as per the reviewer’s comment.

  • Comment: The table highlighting clinical studies is nice. Again, an inhaled azacitidine study is missing (Cheng et al 2021, Lung Cancer).
  • Response: Both studies about Azacitidine and Bevacizumab have been added to Table 3

Reviewer 4 Report

Comments and Suggestions for Authors

The present version of the manuscript is satisfactory and can be accepted as it is.

Author Response

We would like to express our gratitude for the reviewer's time and consideration in reviewing our submitted review

Round 3

Reviewer 2 Report

Comments and Suggestions for Authors
  • Mostly a few minor comments this time. I found quite a few typos and minor factual inaccuracies. The authors are encouraged to check the manuscript again, as these revisions have been turned around very quickly.
  • I still feel there is less discussion of the content in Table 1 than is warranted, and would better help flesh out the paper. The authors should highlight particularly interesting studies from table 1 with discussion in the text.
  • Throughout table 1, "aerosol" and the incorrectly spelled "aerosole" are both used.
  • The bevacizumab study must be removed from Table 3...the work cited is not a clinical trial.

The inclusion of new discussion points on inhaled therapies of each therapeutic class is improved (section 11.2). Please ensure that all of these new references are explicitly stated as being delivered by the inhalation route, rather than systemic. This has been done for some, but not all of the references. These references also don't appear to be in Table 1, and they should be, as they are "examples of current pulmonary-delivered medicines investigated for lung cancer."

Last round, I gave the comment below and received the response:

  • Comment: The idea of highlighting which drugs are repurposed approved drugs in Table 2 is nice. However, many of the drugs in table 1 are also examples of repurposed drugs, they just aren't included in table 2. Perhaps you could combine the tables and have a column highlighting which of the actives are already approved, both as cancer drugs or for other indications.
  • Response: All repurposed drugs have been removed from Table 1 and added to Table 2 in order to organize the data in the tables as per the reviewer’s comment.
  • Second response: The point of the original suggestion was not to remove repurposed drugs from Table 1, but to combine them into a single table to avoid redundancy. As it stands, many more of the drugs in Table 1 are in fact "repurposed" than what is currently in Table 2.

In the inhaled chemotherapy section at the end, it would be good to cite two recent reviews on this topic: Cancers | Free Full-Text | The Position of Inhaled Chemotherapy in the Care of Patients with Lung Tumors: Clinical Feasibility and Indications According to Recent Pharmaceutical Progresses (mdpi.com)

Full article: Inhaled cytotoxic chemotherapy: clinical challenges, recent developments, and future prospects (tandfonline.com)

Author Response

Comments and Suggestions for Authors

  • Mostly a few minor comments this time. I found quite a few typos and minor factual inaccuracies. The authors are encouraged to check the manuscript again, as these revisions have been turned around very quickly.

We express our gratitude to the reviewer for providing their insightful comments. We have thoroughly reviewed the manuscript and made significant modifications to enhance its clarity and accuracy. Please excuse our quick turnaround time, as we are eager to publish this timely work and increase its impact in the field

  • I still feel there is less discussion of the content in Table 1 than is warranted, and would better help flesh out the paper. The authors should highlight particularly interesting studies from table 1 with discussion in the text.

Thank you for providing us with your valuable feedback. We have taken it into serious consideration and made the necessary updates to our report. Specifically, we have included a detailed discussion section that provides a comprehensive analysis of the findings presented in Table 1. In addition, we have highlighted several studies that we believe will be of particular interest to readers who are looking to explore deeper into the topic. We appreciate your input and hope that these changes will enhance the readability and usefulness of our review.

  • Throughout table 1, "aerosol" and the incorrectly spelled "aerosole" are both used.

Thank you for providing us with your valuable feedback. We have corrected this typo throughout the review.

  • The bevacizumab study must be removed from Table 3...the work cited is not a clinical trial.

Thank you for providing us with your valuable feedback. Change made as per reviewer suggestion.

The inclusion of new discussion points on inhaled therapies of each therapeutic class is improved (section 11.2). Please ensure that all of these new references are explicitly stated as being delivered by the inhalation route, rather than systemic. This has been done for some, but not all of the references. These references also don't appear to be in Table 1, and they should be, as they are "examples of current pulmonary-delivered medicines investigated for lung cancer."

We appreciate your valuable feedback. After a thorough review, we have made the necessary changes to the references to ensure that they all pertain to inhaled drugs, while also removing any extraneous studies. Furthermore, we have highlighted the relevant studies and incorporated all of them into Table 1, as per your suggestion.

Last round, I gave the comment below and received the response:

  • Comment: The idea of highlighting which drugs are repurposed approved drugs in Table 2 is nice. However, many of the drugs in table 1 are also examples of repurposed drugs, they just aren't included in table 2. Perhaps you could combine the tables and have a column highlighting which of the actives are already approved, both as cancer drugs or for other indications.
  • Response: All repurposed drugs have been removed from Table 1 and added to Table 2 in order to organize the data in the tables as per the reviewer’s comment.
  • Second response: The point of the original suggestion was not to remove repurposed drugs from Table 1, but to combine them into a single table to avoid redundancy. As it stands, many more of the drugs in Table 1 are in fact "repurposed" than what is currently in Table 2. 

Thank you for providing us with your valuable feedback. We have taken it into serious consideration and made the necessary updates to our report. Specifically, we have included a detailed discussion section that provides a comprehensive analysis of the findings presented in Table 1. In addition, we have highlighted several studies that we believe will be of particular interest to readers who are looking to explore deeper into the topic. We appreciate your input and hope that these changes will enhance the readability and usefulness of our review.

In the inhaled chemotherapy section at the end, it would be good to cite two recent reviews on this topic: Cancers | Free Full-Text | The Position of Inhaled Chemotherapy in the Care of Patients with Lung Tumors: Clinical Feasibility and Indications According to Recent Pharmaceutical Progresses (mdpi.com)

Full article: Inhaled cytotoxic chemotherapy: clinical challenges, recent developments, and future prospects (tandfonline.com)

Thank you for providing us with your valuable feedback. We have incorporated both reviews, as per your suggestions. Thank you again for your time and input.